# Protocell arrays for simultaneous detection of diverse analytes

Yan Zhang [1], Taisuke Kojima[2], Ge-Ah Kim[3], Monica P. McNerney[1,4], Shuichi Takayama [2✉] &
Mark P. Styczynski [1✉]

Simultaneous detection of multiple analytes from a single sample (multiplexing), particularly when done at the point of need, can guide complex decision-making without increasing the required sample volume or cost per test. Despite recent advances, multiplexed analyte sensing still typically faces the critical limitation of measuring only one type of molecule (e.g., small molecules or nucleic acids) per assay platform. Here, we address this bottleneck with a customizable platform that integrates cell-free expression (CFE) with a polymer-based aqueous two-phase system (ATPS), producing membrane-less protocells containing transcription and translation machinery used for detection. We show that multiple protocells, each performing a distinct sensing reaction, can be arrayed in the same microwell to detect chemically diverse targets from the same sample. Furthermore, these protocell arrays are compatible with human biofluids, maintain function after lyophilization and rehydration, and can produce visually interpretable readouts, illustrating this platform's potential as a minimal-equipment, field-deployable, multi-analyte detection tool.

[1] School of Chemical & Biomolecular Engineering, Georgia Institute of Technology, Atlanta, GA, USA. [2] Department of Biomedical Engineering, Georgia Institute of Technology, Atlanta, GA, USA. [3] School of Materials Science and Engineering, Georgia Institute of Technology, Atlanta, GA, USA. [4] Present address: Department of Systems Biology, Harvard Medical School, Boston, MA, USA. ✉email: takayama@gatech.edu; mark.styczynski@chbe.gatech.edu

The ability to sense and characterize analyte concentrations is key to many engineering and biomedical advances, especially when these measurements can be made at or near the point of need[1–7]. While this type of analysis is sometimes straightforward to perform on a single target, many samples with industrial or biomedical relevance have complex analyte profiles and often require measurement of multiple targets spanning different molecular classes (from ions and small molecules to nucleic acids and proteins)[8]. Current sensing and diagnostic tools, especially those designed for use at the point of need, typically only detect one type of analyte at a time, which places significant limitations on the number and the type of tests that can be done on each sample[1–6]. Thus, simultaneously detecting multiple analytes across different molecular classes from a single sample at the point-of-need can empower researchers and clinicians with more information without requiring additional assay time, sample volume, or cost per test.

Unlike abiotic sensors and diagnostics, cells are equipped with extensive capabilities to detect diverse types of analytes simultaneously. Recently developed cell-free biosensors, where cell lysates are used as rich reaction mixtures, have taken advantage of some of this cellular machinery to sense and respond to target analytes via protein or RNA expression[9]. The component reagents can also be lyophilized to enable wide-scale, affordable, and deployable testing[9]. Sensors based on cell-free expression (CFE) have been used for point-of-need testing of environmental contaminants for problems including water quality monitoring[2,10,11]. CFE-based sensors have also been used to detect clinically relevant targets with minimal equipment and operator requirements[3,5], making them excellent candidates for widespread use. However, nearly all current CFE sensors detect only one analyte, even though the sample assayed may contain multiple relevant analytes. As a result, detection of all relevant analytes would require multiple CFE reactions to be run in parallel using multiple aliquots of the sample, introducing handling challenges for onsite testing and collection challenges for samples that are often only available in small volumes. Existing strategies for simultaneous detection of different analytes from a single sample (multiplexing) often require a library of orthogonal reporters, cannot measure analytes from diverse molecular classes (i.e., nucleic acid sequences and small molecules), or cannot be used in minimal-equipment settings[1,6,12–15]. To effectively and efficiently address impactful problems such as the diagnosis of diseases with complex biomarker profiles at the point of need, it is critical to develop improved platforms for simultaneous measurement of analytes across molecular classes.

Arrays of membrane-less protocell sensors formed by polymer-based aqueous two-phase systems (ATPS) and CFE reactions have the potential to address current limitations in multiplexed analyte measurement at the point of need. Mixing two immiscible aqueous polymer solutions can lead to spontaneous liquid–liquid phase separation above certain polymer concentrations[16]. When the two polymers are mixed in appropriate ratios, the phase separation can yield droplets in a bulk phase (Supplementary Fig. 2a)[17–19]. Adding biological machinery (like that of a CFE reaction) to these droplets yields what is essentially a membrane-less precursor of a cell (a protocell)[20,21] with genetic information and the capability to execute complex functions contained in a small, localized volume. Since many soluble macromolecules, such as the proteins and nucleic acids that enable CFE-based sensing, selectively partition from a relatively more hydrophobic polyethylene glycol (PEG)-rich phase to a hydrophilic dextran- or Ficoll-rich phase[16,20,21], the core of the distinct biosensing machinery stays compartmentalized in each ATPS-formed protocell, driven by thermodynamic forces. Because these protocells are membrane-less, this approach also resolves challenges in transporting macromolecular analytes from the bulk phase to

inside the protocells, a limitation present in most membrane-based approaches[22]. For simultaneous detection of target analytes in a single sample, simple topographical placement of micro-basins inside a microwell can be used for stable positioning of distinct protocells[17,18,23] each containing a different CFE sensor (Fig. 1 and Supplementary Fig. 1). Addition of a sample solution to the microwell initiates analyte uptake and compartmentalized detection reactions in multiple isolated membrane-less protocells.

To date, simple protocells formed by PEG and dextran or coacervates have been successfully used to study membrane-less compartmentalization of cell lysates and biomolecules[20,21,24]. Recent studies have also shown that ATPS droplets comprised of dehydrated polymers, including proteins, are compatible with storage at ambient temperature[19,25], enabling low-cost distribution of ATPS immunoassay-based diagnostics to the point of need without cold chain storage and shipping. We envision that an array of membrane-less protocell sensors formed by selective compartmentalization of CFE reactions in ATPS can facilitate simultaneous detection of multiple analytes beyond just proteins, leading to a new class of protocell array-based diagnostics that reports on diverse types of analytes, has a high degree of sensor customizability, and can be used at the point of need.

Toward this goal, here we first assess CFE transcription and translation capabilities in two protocell-forming polymer ATPS (PEG-Ficoll and PEG-dextran) with an in-house prepared *E. coli* lysate. We show that analytes (small molecules and nucleic acids) added to a microwell containing an array of CFE protocells selectively activate their cognate sensors confined to distinct protocells with comparable sensitivity to those in nonprotocell CFE reactions. We highlight the utility of this platform for simultaneous, multimodal biomarker detection via the detection of clinically relevant targets (e.g., nucleic acids from pathogenic bacteria and micronutrients) from the same sample and in a human serum matrix. Only one reporter (green fluorescent protein, GFP) is needed for multiplexed analyte measurement, which

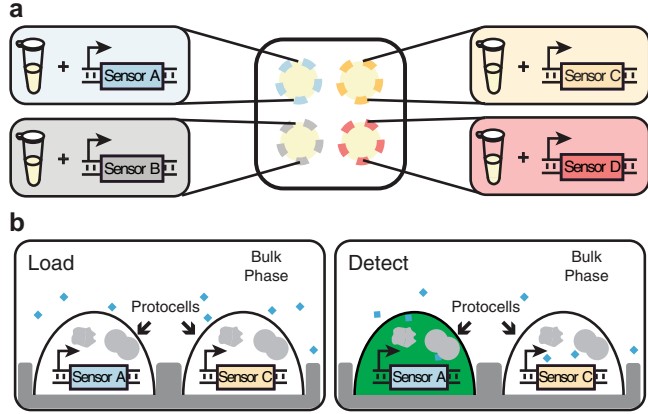

**Fig. 1 Schematic of the membrane-less protocell array platform for simultaneous detection of multiple analytes.** The use of a spatially patterned array of phase-separated protocell sensors enables each microwell to provide measurements for multiple analytes using a single fluorescent or colorimetric reporter. **a** An overhead view of a microwell with four micro-basins. White area surrounding the micro-basins (yellow-shaded regions) represents the bulk phase formed by a PEG solution containing the analytes to be detected. Each micro-basin has a protocell containing a CFE reaction (represented by the microcentrifuge tube) with a different sensor plasmid or genetic circuit. **b** Side view of a microwell with micro-basins. Analytes in the bulk phase (blue diamonds) enter protocells to activate their cognate sensor (blue DNA construct), producing detectable reporter signals. Multiple analytes can be measured in parallel by tracking which protocell produces reporter signals (green color on the right).

significantly reduces the complexity of test development and reconfiguration. Protocell arrays also meet key criteria for field-deployable sensors and diagnostics: we demonstrate that GFP reporters can be easily replaced with color-producing enzymatic reporters to enable equipment-free test interpretation, and we demonstrate parallel, simultaneous detection of multiple analytes using protocell arrays that have been lyophilized for storage at ambient temperature. Taken together, the presented protocell array platform both expands the reach of cell-free biosensors and provides the foundation to develop advanced, customizable diagnostic chips for simultaneous detection of diverse types of analytes at the point of need.

## Results

**Compartmentalization of cell-free protein expression in ATPS-formed protocells**. While CFE-based protocells have previously been described[20,24], their application to analyte detection has not. To use membrane-less CFE protocells for simultaneous measurement of multiple analytes, the sensor plasmids and core machinery for transcription and translation must retain activity in an ATPS environment and stay compartmentalized inside the protocell. We first verified that CFE reactions based on an in-house prepared *E. coli* lysate maintain transcription and translation function in the context of an ATPS. CFE reactions

constitutively producing GFP were tested in two ATPS environments: a 5% 35k PEG–5% 500k dextran system and a 5% 35k PEG–10% 400k Ficoll system. These pairs of polymers were selected based on their ability to separate into two liquid phases without pre-equilibration to create a metastable system with mass transfer enhancements from convective mixing[25], our past experience indicating their stability in dehydrated ATPS systems[19,25], and previously reported evidence that these polymers are compatible with CFE machinery[20,24,26]. Polymer concentrations required to form biphasic separation were selected based on previously characterized binodal curves[25,27] as well as our own tests of their compatibility with in-house prepared CFE lysate and reagents (Supplementary Fig. 2b, c). Additional rheological data are also available in the literature[25,27]. PEG constitutes the bulk phase for both ATPS environments, while dextran or Ficoll constitutes the protocell (Fig. 2a). Individual protocells were first formed by mixing CFE reactions with Ficoll or dextran polymer until homogenous and pipetting a 2 µL droplet into a bulk phase PEG solution. No protein production was observed after 3 h of incubation at 37 °C (Fig. 2b), perhaps due to the necessary salts and building blocks for protein synthesis (nucleotides and amino acids originally in the protocell) not remaining strongly partitioned in the protocell but instead diffusing into the bulk

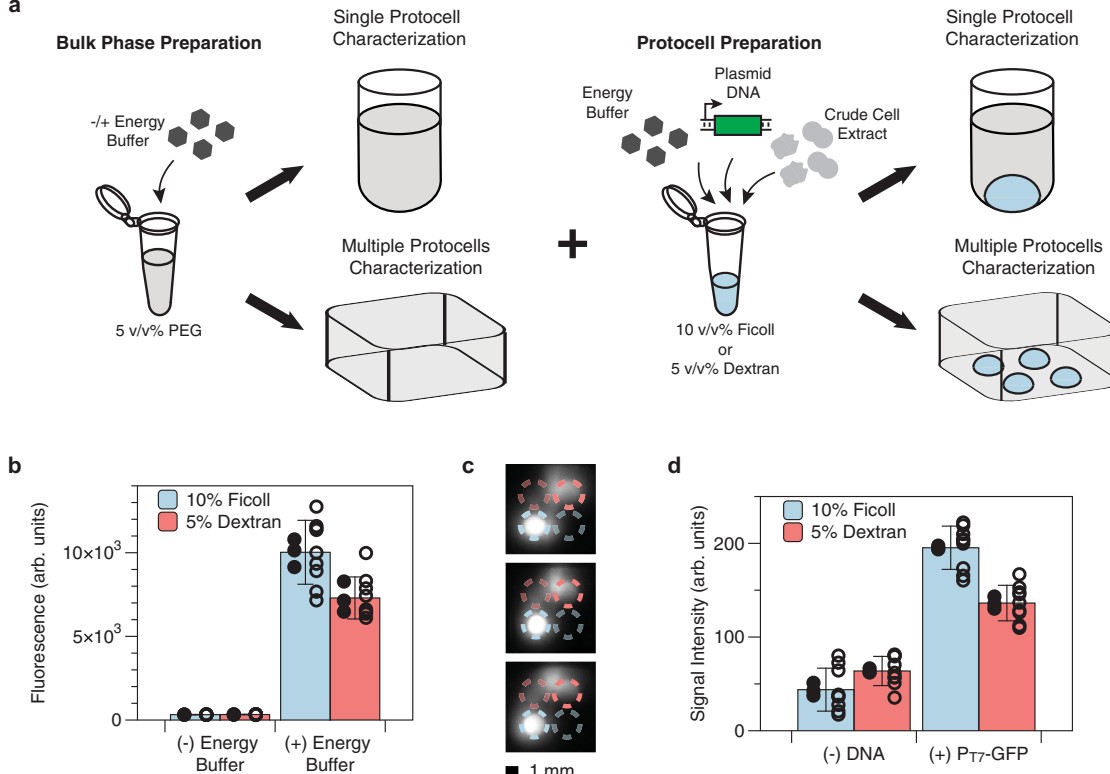

**Fig. 2 Characterization of protein expression and reaction compartmentalization in membrane-less protocells. a** Schematic of single-well and multi-well protocell reactions and their preparation methods. A small volume of Ficoll or dextran solution containing reagents for cell-free expression of GFP was pipetted into a bulk phase solution of PEG to form a protocell. Subsequent reactions were incubated at 37 °C for 3 h. **b** Fluorescence of Ficoll and dextran protocell reactions in single-protocell format were measured with a plate reader with 485/528 nm excitation/emission wavelengths. For detectable protein expression, energy buffer must be supplemented in the bulk phase. **c** Fluorescence image of protocell arrays obtained by a ChemiDoc MP imager (0.5 s exposure time, 530/28 nm filter) for three biological replicates. Contents of individual protocells are indicated with colored circles: blue for Ficoll polymer-encapsulated reactions, red for dextran-encapsulated reactions. Bright colors indicate CFE reactions containing GFP plasmid for expression, and faded colors indicate negative controls with no plasmid. Scale bar is 1 mm. **d** Quantitative assessment of fluorescence image in **c**, with pixel intensity quantified by image processing software (Fiji). Data are presented as mean values ± SD of 9 replicates (3 biological replicates × 3 technical replicates). Solid-filled circles represent the mean of each biological triplicate, and hollow circles represent all data points. Details on CFE lysate used and plasmid concentrations are provided in Supplementary Table 1. Note, due to different measurement instruments being used for **b** and **d**, the arbitrary units on the y-axes are different scales.

phase and thus diluting their concentrations. Supplementing CFE energy buffer in the PEG bulk phase enabled CFE protein synthesis (Fig. 2b).

We next validated that ATPS-formed protocells compartmentalize multiple CFE reactions, such that reporter plasmids and proteins remain in their phase-separated droplets in a membrane-less protocell array. We performed CFE reactions in a custom-designed 96-well plate where each microwell contains four micro-basins for protocell placement (Supplementary Fig. 1a), with the bulk phase supplemented with energy buffer. CFE reactions with or without a plasmid coding for GFP were combined with dextran or Ficoll, loaded into the micro-basins (Fig. 2a), and incubated at 37 °C for 3 h. A fluorescence imager (ChemiDoc MP) was used to measure protein production in the protocell array in this custom microwell plate. GFP-producing Ficoll protocells were better localized in their designated micro-basins compared to those of dextran protocells (Fig. 2c, d), demonstrating that cell-free lysate, plasmids, and reporter proteins are better compartmentalized in the PEG-Ficoll environment. This result also suggests that multiple protocells expressing different biosensors could be arrayed in the same microwell, enabling a platform for parallel, simultaneous detection of multiple targets. We note that GFP diffusion from high GFP-producing protocells is visible in the fluorescence images at later time points for both PEG-dextran and PEG-Ficoll systems (Supplementary Fig. 2d, e). Since typical cell-free sensing reactions are on the order of 0.5–3 h, such issues at later time points in PEG-Ficoll protocells should be less problematic for detection applications, although some degree of nonspecific crosstalk due to these limitations is possible.

**Multiplexed detection of model small molecule and nucleic acid systems.** To demonstrate that these membrane-less CFE protocell arrays can be used for simultaneous detection of multiple analytes, we incorporated sensors that respond to multiple model small molecules. Plasmids encoding each sensor system were embedded in different protocells, with analytes added to the bulk phase to diffuse into the protocells to activate their cognate sensor proteins. To evaluate small molecule detection, we tested an iso-propyl β-D-1-thiogalactopyranoside (IPTG)-inducible LacI-$P_{T7lacO}$ circuit and an arabinose-inducible AraC-$P_{BAD}$ circuit (Fig. 3a)[28]. In each biosensing circuit, the small molecule inducer modulates the activity of its corresponding transcriptional regulator (IPTG dere-presses LacI and arabinose activates AraC), allowing transcription of GFP reporter from its respective promoter ($P_{T7lacO}$ or $P_{BAD}$). The arabinose CFE sensor uses a single plasmid encoding constitutive AraC expression and $P_{BAD}$-regulated GFP expression from the same divergent promoter. The IPTG sensor uses one plasmid to express GFP from a $P_{T7lacO}$ promoter and another to express LacI constitutively. When IPTG and arabinose were individually added to the bulk phase, only the appropriate sensor reactions were activated, with minimal signal crosstalk (Fig. 3b). This result demonstrates that protocell arrays can successfully detect multiple small molecule signals simultaneously.

We next demonstrated that protocell arrays could also enable simultaneous detection of multiple RNA targets using previously reported model toehold switches to control the output of CFE reactions[7]. A toehold switch is a *de novo* designed RNA regulator that forms an inhibitory hairpin to prevent translation of a downstream protein[29]. Addition of a trigger RNA with sequence complementarity to part of the switch unfolds the hairpin, turning on reporter expression (Fig. 4a). We constructed plasmids in which two previously characterized orthogonal toehold switches (B and H)[7] are constitutively expressed from a T7 promoter, and used these sensor plasmids in the protocell array (Fig. 4b). GFP expression for a given protocell sensor was

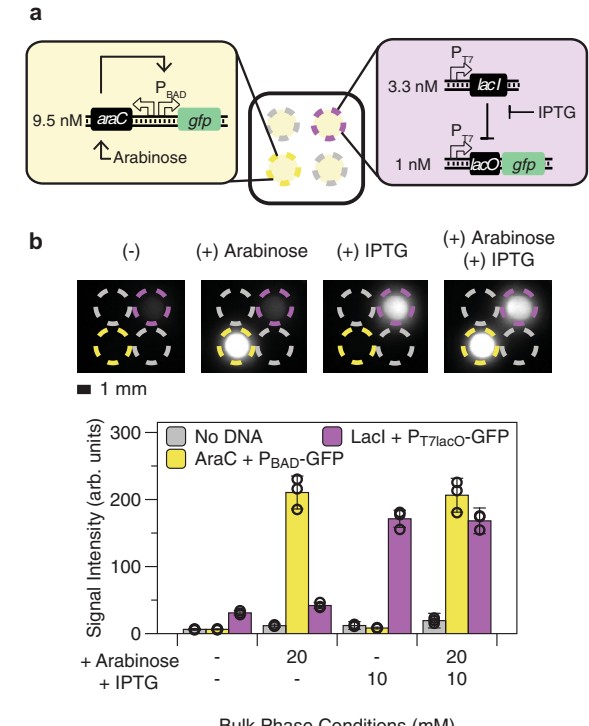

**Fig. 3 Simultaneous detection of multiple model small molecules in membrane-less protocell arrays. a** Schematic of protocell array setup for simultaneous detection of two small molecules. Yellow and purple circles indicate micro-basins containing arabinose and IPTG sensors, respectively. Gray circles are micro-basins containing CFE protocells without plasmids. The circuit diagrams and plasmid concentrations used for the arabinose and IPTG sensors are shown in the insets. **b** Representative fluorescence images and their quantification results after 3 h of incubation at 37 °C. Small molecules added to the bulk phase for each experiment are indicated above each image. Circle colors are as in **a**. Each protocell sensor is only activated when its cognate small molecule (10 mM IPTG, 20 mM arabinose, or both) is present in the bulk phase. Data are presented as mean values ± SD of 9 replicates (3 biological replicates × 3 technical replicates) for protocell sensors and 18 replicates for controls with no plasmid (3 biological replicates × 6 technical replicates). Hollow circles represent the mean of each biological triplicate. Scale bar is 1 mm. Data on each biological triplicate are provided in Supplementary Fig. 3. Details of CFE lysate used and plasmid sensor concentrations are provided in Supplementary Table 1.

only observed when the cognate RNA trigger for the switch in that protocell was present in the bulk phase, demonstrating that protocell arrays can successfully detect multiple RNA targets in parallel (Fig. 4c, d).

Since DNA is more stable than RNA and can serve as a template for producing many RNA molecules, we hypothesized that protocell sensors could respond more sensitively to triggers expressed from a DNA template. RNA can be readily converted into DNA using reverse-transcriptase mediated isothermal amplification techniques like Nucleic Acid Sequence-Based Amplification (NASBA) or Reverse Transcription Recombinase Polymerase Amplification (RT-RPA)[30,31], making the detection of DNA-templated RNA a viable strategy for field-deployable detection of RNA targets. Just 0.2 nM of linear DNA (expressing RNA under the control of a T7 promoter) added to the bulk phase strongly activated GFP expression, with slight activation detectable upon adding just 20 pM of linear DNA (Fig. 4e, f). Interfacing CFE sensors with protocell arrays does not compro-mise sensor sensitivity. Toehold switches in both single-phase

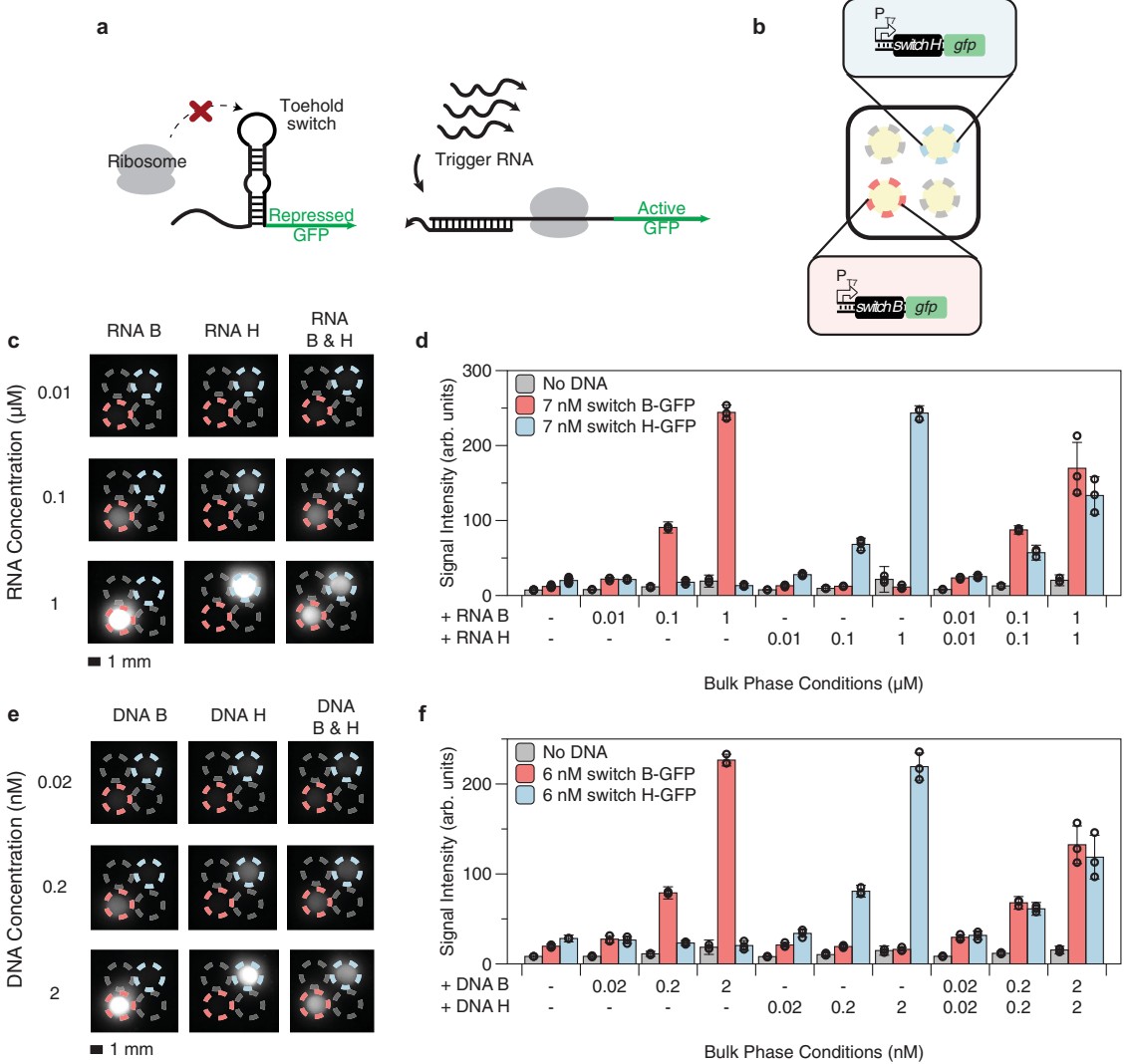

**Fig. 4 Simultaneous detection of multiple model nucleic acid sequences in membrane-less protocell arrays.** Reactions were incubated at 37 °C for 3 h. **a** Schematic of toehold switch mechanism. Without trigger RNA, the switch mRNA forms an inhibitory hairpin blocking the translation of a reporter (GFP). Addition of trigger RNA unwinds the switch hairpin, allowing GFP translation. **b** Schematic of protocell array setup for simultaneous detection of multiple nucleic acid sequences. Red and blue circles indicate micro-basins containing toehold switches B and H, respectively. Gray circles indicate CFE protocells without plasmids. **c** Representative fluorescence image of RNA detection from 10 nM to 1 µM. Images in the same column have the same RNA trigger(s) added. Images in the same row have the same concentration of trigger(s) added. Circle colors are as shown in **b**. **d** Quantification of fluorescence images in **c** and their replicates. Addition of both RNA triggers mutually represses their outputs, but this effect is specific to these triggers. **e** Representative fluorescence image of linear DNA detection from 20 pM to 2 nM. Images in the same column have the same DNA trigger(s) added. Images in the same row have the same concentration of DNA trigger(s) added. Circle colors are as shown in **b**. Addition of both DNA triggers also mutually represses their outputs. **f** Quantification of fluorescence images in **e** and their replicates. For experiments where the bulk phase had no triggers, data are presented as mean values ± SD of 27 replicates for protocells containing toehold switch sensors (9 biological replicates × 3 technical replicates) and 54 replicates for reactions with no DNA (9 biological replicates × 6 technical replicates). For experiments with triggers (RNA or DNA), data are presented as mean values ± SD of 9 replicates for protocells containing toehold switch sensors (3 biological replicates × 3 technical replicates) and 18 replicates for reactions with no DNA (3 biological replicates × 6 technical replicates). Hollow circles represent the mean of each biological replicate. Scale bar is 1 mm. Data for each biological replicate are provided in Supplementary Figs. 4 and 5. Details on CFE lysate used, plasmid concentrations, and reaction additives are provided in Supplementary Table 1.

CFE and protocell arrays showed similar limits of detection of approximately 10 nM for RNA triggers (Supplementary Fig. 6) and 20 pM for DNA triggers (Supplementary Fig. 7). While the simultaneous presence of both triggers B and H caused some suppression of both signals compared to either one alone (Fig. 4d, f), this effect was also observed in single-phase CFE reactions and was found to be specific only to triggers B and H (Supplementary Fig. 8), suggesting adverse basepair interactions among trigger sequences that are independent of protocell array-based sensing.

**Multimodal detection of clinically relevant targets in a human serum matrix.** Having demonstrated the multiplexing capabilities of protocell arrays, we next used this platform to detect multiple types of clinically relevant biomarker molecules with public health relevance: ions/minerals, small molecules, RNA, and DNA. The ions and small molecules we chose for detection were two micronutrients, zinc and vitamin $B_{12}$ (adenosyl-cobalamin), that are important sensing targets for global health applications and for which our group has previous experience developing CFE

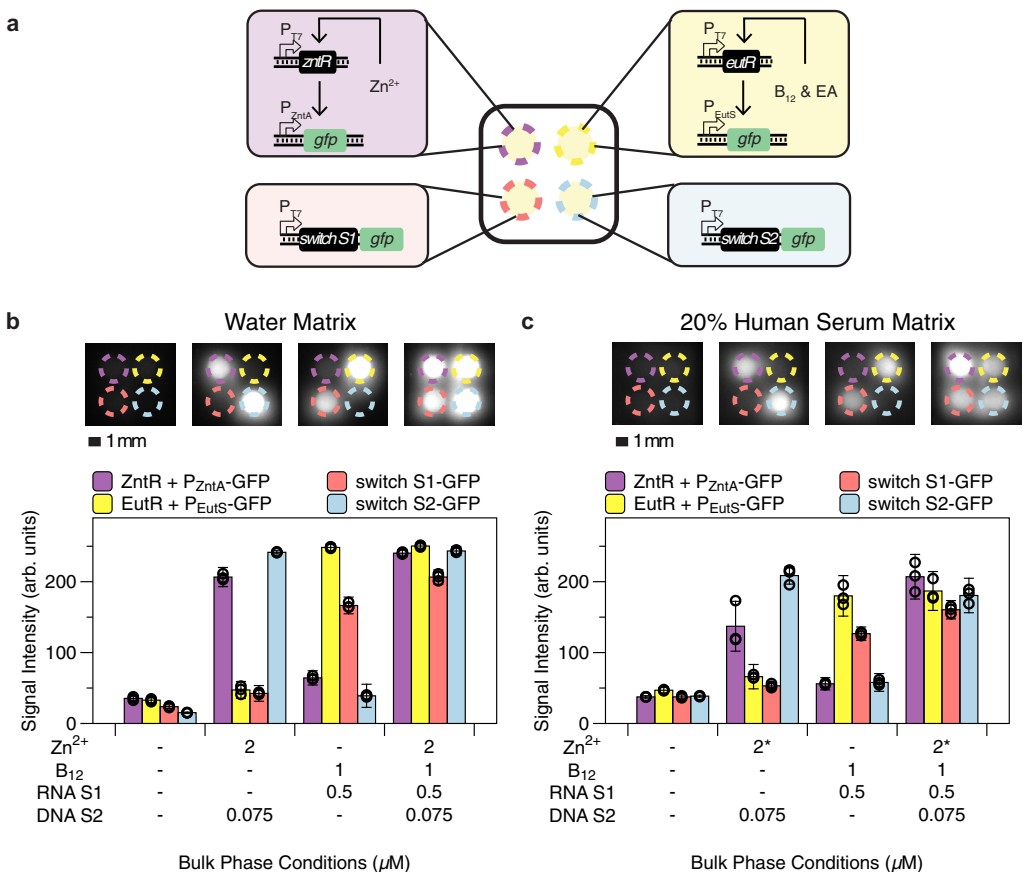

**Fig. 5 Simultaneous detection of multiple clinically relevant biomarkers across multiple molecular classes in water and 20% human serum matrices.**
Reactions were incubated at 37 °C for 3 h. **a** Schematic of membrane-less protocell array setup for multimodal, simultaneous detection of diverse classes of clinically relevant biomarkers. Purple and yellow circles indicate micro-basins containing zinc and vitamin B_{12} sensors, respectively. Red and blue circles indicate micro-basins containing Stx1 (S1) and Stx2 (S2) toehold switches, respectively. **b** Representative fluorescence image (top) and quantification of the representative image and its replicates (bottom), demonstrating multimodal target measurement using protocell arrays with simultaneous detection of ion, small molecule, RNA, and DNA targets from the same bulk phase. Data for each biological replicate are provided in Supplementary Fig. 11. **c** Multimodal target detection in a 20% human serum matrix. Representative fluorescence image (top) and quantification of the representative image and its replicates (bottom), demonstrating the robustness of protocell arrays to a complex sample matrix. The asterisk (*) next to the zinc concentration indicates the total zinc concentration is 2 µM in the bulk phase after accounting for the zinc remaining in chelated human serum (Supplementary Fig. 14). Of note, the sensor plasmid concentrations were increased in **c** compared to **b** to yield comparable output signal; this tuning was necessary due to the decrease in protein expression that occurs in human serum[5]. Data for each biological replicate are provided in Supplementary Fig. 13. Data are presented as mean values ± SD of 9 replicates (3 biological replicates × 3 technical replicates). Hollow circles indicate the mean of each biological replicate. Scale bar is 1 mm. Details on CFE lysates used, plasmid concentrations, and reaction additives are provided in Supplementary Table 1.

biosensors[4,5]. The zinc sensor uses the activator ZntR, which activates expression from its cognate promoter P_{ZntA} when zinc is present. The B_{12} sensor uses the activator EutR, which activates expression from its cognate promoter P_{EutS} when both B_{12} and the cofactor ethanolamine (EA) are present. We characterized these micronutrient biosensors together in protocell arrays to confirm sensor performance (Supplementary Fig. 9).

For nucleic acid detection, we developed and used toehold switch sensors that detect nucleic acid sequences from two bacterial pathogens: Shiga toxin-producing *E. coli* (STEC), which is a foodborne pathogen that causes life-threatening gastrointestinal symptoms[32], and *Bacteroides thetaiotaomicron* (*B. theta*), which is linked to increased virulence of STEC[33]. We used a previously developed toehold switch for *B. theta*[34] but had to design new switches for STEC, as previously published *E. coli* switches would cross-react with nonpathogenic *E. coli* strains[34]. We developed two STEC switches targeting genomic sequences of toxin proteins: Shiga toxin I (Stx1) and Shiga toxin II (Stx2)[35]. When different combinations of linear DNA coding for *B. theta*,

Stx1, and Stx2 triggers amplified from the genomic DNA of *B. theta* and STEC O157:H7 were added to the bulk phase of a microwell containing protocell arrays, all protocell sensors showed orthogonal trigger detection with minimal reaction crosstalk (Supplementary Fig. 10), demonstrating that protocell arrays can reliably detect multiple pathogenic bacteria nucleic acid sequences in parallel.

We then combined the validated micronutrient and bacterial sensors to demonstrate multimodal detection of diverse classes of analytes in a chemically defined sample and a contrived human serum sample. A protocell array containing sensors for zinc, B_{12}, Stx1, and Stx2 was deposited to the microwell along with different combinations of analytes (zinc, B_{12}, Stx1 trigger RNA, and linear DNA for Stx2 trigger expression) in the bulk phase (Fig. 5a). Each protocell sensor produced GFP only when its cognate analyte was present, demonstrating successful multiplexed detection spanning multiple molecular classes (ion, small molecule, RNA, and DNA) from a single sample (Fig. 5b). Notably, the protocells used for micronutrient and bacterial sensors compartmentalized different

CFE lysates—micronutrient sensors used *E. coli* lysate with basal level T7 RNAP and bacterial sensors used lysate enriched in T7 RNAP (see Methods and Supplementary Table 1). The Stx2-sensing protocell also contained 2 μM χDNA to protect Stx2 trigger DNA from exonuclease degradation by *E. coli* lysate once it entered the protocell to activate reporter production[36]. Because other protocells did not have χDNA, the competition for transcriptional machinery from Stx2 trigger that had diffused into those protocells was thus minimized, as that linear DNA would get rapidly degraded in those protocells[36]. This showcases the fact that multiple customized sensing reactions can coexist in the same protocell array without negatively impacting neighboring protocell reactions.

With this multimodal, multiplexed analyte sensing platform established, we next sought to verify that detection with protocell arrays remains robust even in complex samples such as human serum. We increased sensor plasmid concentrations to compensate for nuclease activity in serum[37,38] and spiked 20% human serum with target molecules at levels similar to those used in our previously published CFE efforts[5]. With these contrived serum samples as the bulk phase, we observed target-specific activation from all protocell sensors (Fig. 5c), demonstrating our platform's translational potential for assessment of biomarkers in patient samples.

We note that the presence of Stx1 RNA and $B_{12}$ in the bulk phase caused a slight increase in overall expression from the zinc sensor (both in basal expression levels in the $+Stx1 + B_{12}$ case compared to the null case, and in activated levels in the +all case compared to the $+Stx2 + Zn^{2+}$ case, Fig. 5 and Supplementary Figs. 11-13). This is a prototypical example of a matrix effect, where analytes other than a sensor's target analyte that are present in the same sampling matrix can cause changes in a sensor's response[39–41]. Matrix effects are a widely-encountered analytical issue that must be addressed for almost any quantitative sensor or diagnostic in a complex sample matrix[5,39–41].

In addition to matrix effects due to different target analytes, the presence of human serum also has inhibitory effects on CFE protein production[5], with the effects being more prominent with increasing serum concentrations (Fig. 5 and Supplementary Figs. 11–13). Serum's negative impact on CFE protein production is potentially caused by nuclease activity[37,38]. As a result, we increased the plasmid concentrations of biosensors to counteract the loss in protein production and provide comparable signal output in a serum matrix (Supplementary Table 1).

**Toward a field-deployable, equipment-free diagnostic platform**. To be useful as a minimal equipment, point-of-care sensing and diagnostic tool, the membrane-less protocell array platform should produce test results that can be visually interpreted without the aid of equipment like plate readers or fluorescence imagers. Toward this goal, we replaced the GFP reporter protein with β-galactosidase (LacZ), an enzyme that catalyzes the production of a colorimetric readout. We tested two colorimetric substrates, chlorophenol red-beta-D-galactopyranoside (CPRG) and 5-Bromo-4-chloro-3-indolyl β-D-galactopyranoside (X-gal), that are substrates for LacZ and are widely used for cell-free diagnostics and molecular biology assays, respectively[3,5,7,34,42]. We also used a new white microwell design to improve pigment visualization, reduce the bulk phase volume requirement, and increase the number of micro-basins in a microwell to include more reactions (Supplementary Fig. 1b). We found that, while LacZ could produce visible color change 30 min faster when cleaving CPRG than when cleaving X-gal, the product of CPRG cleavage rapidly diffused from the protocell to the surrounding bulk phase (Supplementary Fig. 15). This diffusion would obscure

result interpretation if readings were not taken within 30 min of color change, which could be an issue if different multiplexed sensors have different characteristic response times. As a result, we chose to use X-gal for subsequent test development based on its stable localization of pigments, despite a longer reaction time.

We next used this colorimetric reporter to detect the presence of nucleic acid sequences characteristic of pathogenic bacteria. A protocell array with toehold switch-based sensors for *B. theta*, Stx1, and Stx2 was loaded into micro-basins, and the bulk phase contained combinations of linear DNA coding for expression of cognate triggers amplified from the genomic DNA of *B. theta* and STEC O157:H7 (Fig. 6a). All protocell sensors turned visibly blue to accurately report on the presence of the triggers within 2 h with minimal pigment leakage to neighboring reactions (Fig. 6b).

Additionally, a field-deployable protocell array-based sensing platform must be compatible with storage and transportation at ambient temperature for easy distribution to the point of need. To demonstrate our system's ability to meet this criterion, we lyophilized arrays of Ficoll CFE solutions containing different bacteria nucleic acid sensors in micro-basins and separately lyophilized the bulk phase solution (containing PEG, X-gal, and cell-free energy buffer). We show that the addition of water- or trigger-reconstituted bulk phase to the array formed the protocells via liquid–liquid phase separation and activated the CFE sensors in the protocell arrays. All bacteria-sensing protocells remained compartmentalized in their micro-basins, produced visible color change within 3 h of incubation at 37 °C, and accurately reported the triggers added to the bulk phase (Fig. 6c). This result suggests the possibility for a simple-to-operate multiplexed assay that can be performed by minimally trained staff, which is critical for point-of-need use in the field of environmental or epidemiological surveillance.

## Discussion

In recent years, membrane-less protocell models have been used for various scientific and engineering applications, including cell-like functionalities such as selective nucleic acid retention[21,43], reaction acceleration[44,45], and droplet division[46]. Despite being membrane-less, these phase-separated droplets are quite robust, maintaining their structure through dehydration and rehydration cycles[47] and enabling potentially impactful downstream uses. One example is the acoustically trapped protocell patterning technique that enables different enzymatic reactions to occur in adjacent protocells[48]. However, such a strategy requires complex equipment and extensive optimization of reaction environments, making it challenging to implement at the point of need.

Our approach using polymer ATPS to construct protocell arrays can address current limitations in bringing multiplexed analyte measurement to field applications. We show that topographical micro-basin features patterned on the floor of standardized 96-well plate format microwells (Supplementary Fig. 1) enable multiple protocells to coexist spatially separated without any external input to maintain separation. The membrane-less aspect of protocells formed by ATPS also facilitates minimally hindered analyte diffusion (and even concentration for some analytes with favorable partitioning behaviors[16,20,49]) into the protocells. Furthermore, these protocells can compartmentalize custom-designed CFE reactions that can detect diverse classes of analytes, remain robust to complex sample environments, and retain their sensing capabilities after lyophilization for on-demand, simultaneous measurement of multiple analytes at the point of need.

The embedding of CFE reactions in ATPS-formed protocells is uniquely poised to meet many key challenges in biosensing applications, as this platform can be customized to detect diverse

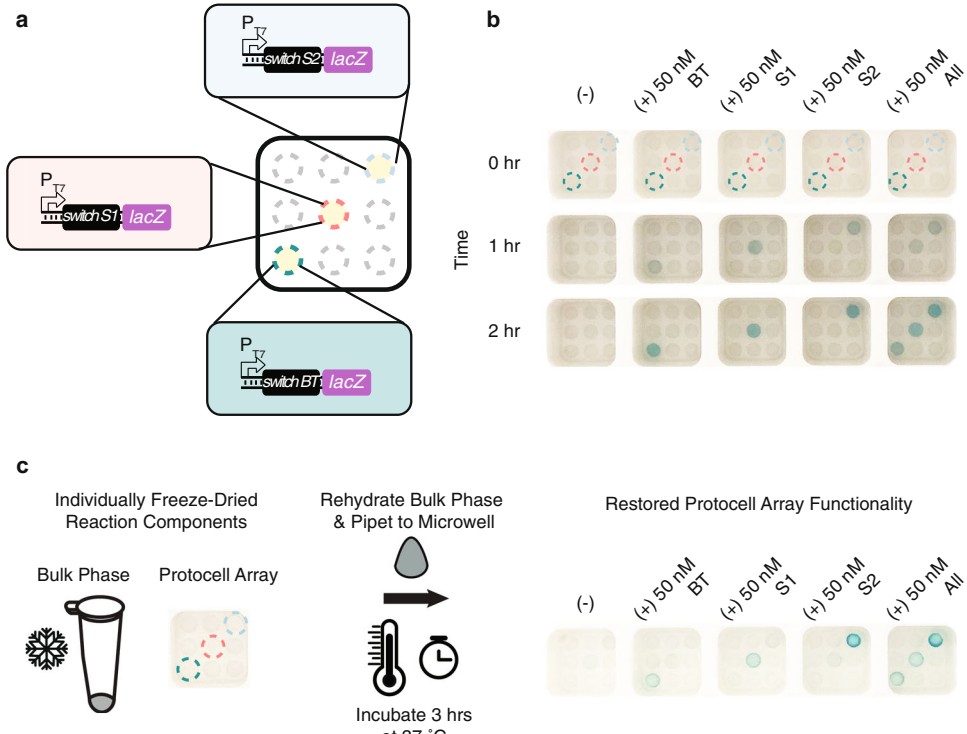

**Fig. 6 A membrane-less protocell array meets key criteria for a minimal equipment, field-deployable multiplexed assay. a** Schematic of colorimetric protocell array setup for simultaneous detection of multiple nucleic acid sequences. Teal, red, and blue circles indicate micro-basins containing *B. theta* (BT), Stx1 (S1), and Stx2 (S2) protocell sensors, respectively. Gray circles indicate empty micro-basins without CFE reactions. **b** Representative sensor activation and pigment production in different bulk phase conditions and at different time points. Images in the same row were taken at the same incubation time. The concentration and the type of DNA trigger(s) used are indicated above each image. Dashed circles indicate micro-basins containing protocell sensors, as shown in **a**. Results from three biological replicates, as well as their technical triplicates, are provided in Supplementary Fig. 16. **c** Protocell array functions after lyophilization. Protocells containing CFE reactions coding for sensors of bacterial nucleic acid sequences were lyophilized in micro-basins within microwells, and bulk phase solutions were lyophilized separately. Addition of sample-reconstituted bulk phase to microwells formed the protocells via liquid–liquid phase separation and revived CFE sensing reactions, leading to pigment production after 3 h of incubation at 37 °C. Three biological replicates, as well as their technical triplicates, are provided in Supplementary Fig. 17.

sets of analytes by reprogramming the DNA sequence of plasmids used as the basis for sensing. Previous work has demonstrated that CFE can be used to detect metabolites via incorporation of genetically encoded metabolic transducers[13,50], proteins via genetically encoded aptamers[51,52], and nucleic acids via genetically encoded synthetic regulators[3,34,53]. These different sensing systems often require differently optimized CFE reactions (e.g., lysate preparations or macromolecular reaction additives), but these individually optimized sub-systems can all be used together in parallel to measure different targets in the same sample. Of note, the protocells sensing nucleic acids in Fig. 5 used a T7 RNAP-enriched lysate, while protocells sensing small molecules did not. Moreover, only the protocell expressing Stx2 switch contained $\chi$DNA for linear DNA protection, which helped to minimize competition for transcription resources in the other protocells as the linear DNA coding for Stx2 trigger would be degraded in the other protocells. Selectively tuning an individual sensor's performance without negatively impacting other sensors is an enabling advance for simultaneous, parallel measurement of multiple analytes, made possible by encapsulating individual sensors into discrete, phase-separated protocells. The compartmentalization of protocell sensors eliminates the need for multiple, orthogonal, or target-specific reporter systems, allowing all sensors in the same microwell to be developed with the same reporter system. This facilitates rapid development of test panels, as newly developed sensors can be readily incorporated into existing protocell arrays with obsolete sensors easily replaced.

An array of spatially separated gene networks that sense and respond to different analytes in the same environment has many potential applications. One such example is in agricultural and environmental surveillance[2], where a small environmental sample can indicate the presence of a pollutant, excessive fertilizer, or even an animal or plant pathogen. Another example would be to study crosstalk between protocells. While interactions between sensor protocells would be construed as a problem for developing diagnostics, the ability for individual protocells to produce small molecules that could diffuse to other protocells in the array could be used as a platform for prototyping chemical communication networks among synthetic cells[54,55]. Each protocell could be separately programmed with a different function, allowing for the investigation of different communication networks at the single-(proto)cell level. In addition, the physical arrangment of protocell arrays—a reaction phase with an interface to a bulk liquid where diffusion of small molecules can occur freely between the two liquids—is similar to a CFE reaction operating in dialysis mode, which has been shown to extend CFE reaction lifetime[56]. In dialysis operation, toxic molecules produced in CFE reactions can diffuse out of the reaction volume, potentially improving CFE yield. Thus, protocells could provide a new way to accrue the benefits of dialysis mode reactions without expensive dialysis membranes or cassettes.

Perhaps the most impactful application of these membrane-less protocell arrays is for point-of-care diagnostics with simultaneous, parallel detection of multiple clinically relevant

biomarkers in human biofluids. Many diseases and disorders that clinicians and researchers seek to identify in the field are not diagnosed based on just one biomarker but by the combination of multiple test results, often spanning multiple classes of biomolecules. The protocell array formed by CFE and ATPS is an enabling platform for these needs. It is compatible with complex sample matrices and withstands lyophilization for intact functionality upon rehydration, enabling flexible test development and deployment to the point of need without expensive cold chain requirements. However, in our efforts to demonstrate serum compatibility with protocell arrays, we found that serum proteins formed cloudy precipitates when added to a PEG-rich bulk phase. While serum protein precipitation did not obstruct biphasic polymer separation and detection of GFP signal, it hindered micro-basin and pigment visualization in colorimetric reactions. One possible way to resolve this might be to use clear-bottomed plates such that pigments can be visualized from the bottom[17].

The results in Fig. 5 demonstrate our platform's translational potential for assessment of biomarkers in patient samples. We achieved zinc detection at 2 μM, which is within the clinically relevant range for zinc deficiency (1.7–2.3 μM in 20% human serum)[5,57]. Unfortunately, there is currently no CFE-based $B_{12}$ sensor with a limit of detection in the clinically relevant range for $B_{12}$ deficiency (15–80 pM in 20% serum)[4,58]. Activation of the $B_{12}$ sensor at 1 μM nonetheless offers a proof-of-concept toward simultaneous detection of diverse classes of analytes in a complex human serum matrix. In addition, since pathogen nucleic acids are typically present at femtomolar concentrations[3,6,34] and current limits of detection of CFE sensors for these targets are typically at the nanomolar level (Supplementary Fig. 10), nucleic acid amplification methods such as NASBA or RT-RPA would need to be integrated into the sample workup or directly into protocell arrays to bring these targets into the sensor response range. To this end, we have demonstrated that protocell arrays are robust to detecting triggers amplified from RPA reactions (Supplementary Fig. 18), but target-specific optimization of primer pairs and reactions would be necessary to achieve parallel amplification of triggers in the same RPA reaction. This input amplification (common to nearly all current CFE nucleic acid sensors[3,14,34,53]) does, however, increase the level of complexity of the test and require trained operators to execute. Fortunately, ongoing research efforts are moving towards eliminating the need for upstream input amplification[59]. Given the demonstrated modularity and easily reconfigurable format of our platform, those resulting amplification-free CFE systems could be incorporated into our protocell array and enable one-pot detection of clinically relevant biomarker levels. Nonetheless, one would still need to ensure that any upstream processing does not alter levels of other targeted biomarkers present in the sample.

In conclusion, we have demonstrated that arrays of polymer ATPS-formed, membrane-less protocells with compartmentalized DNA transcription and RNA translation machinery can perform simultaneous detection of diverse classes of analytes. Such a protocell system survives lyophilization to enable test storage and distribution at ambient temperature. Rehydration with an analyte-containing polymer solution reestablishes phase separation, allows uptake of analytes by the resulting protocell sensors, and revives compartmentalized transcription- and translation-mediated detection reactions. These arrays of membrane-less protocells provide modular, field-deployable, multimodal, multiplexed diagnostic potential. Interfacing CFE reactions with membrane-less protocells addresses the current limitation in simultaneous detection of diverse analytes from a single sample and opens new opportunities for implementing diagnostic panels for use at the point of care.

## Methods

**Bacterial strains.** *E. coli* strain DH10β was used for all cloning and plasmid preparations. *E. coli* strain BL21 Star (DE3) Δ*lacIZYA* was created by lambda red recombination[60] and used for in-house cell-free lysate preparation. Genomic DNA from *E. coli* strains DH10β and BL21 Star (DE3) were used as negative controls for target-specific amplification of Stx1 and Stx2 triggers (Supplementary Fig. 19). Genomic DNA from *B. thetaiotaomicron* (ATCC 29148D) and STEC O157:H7 (ATCC 51657GFP) were used for testing detection of pathogenic bacteria.

**Genetic parts assembly and plasmid preparation.** Sequences of all parts used in this study are provided in the Source Data, under file name Sequence Information. DNA oligonucleotides for cloning and sequencing were synthesized by Eurofins Genomics. Partial sequences for small molecule sensors and toehold switches were obtained from previously published sequences and were synthesized either as gene fragments or ssDNA-annealed oligonucleotides from Eurofins Genomics. Plasmids expressing regulators and reporter proteins were cloned using either Gibson Assembly[61] or blunt end ligation into plasmid backbone pJL1. Assembled constructs were transformed into DH10β cells, and isolated colonies were grown overnight in LB with antibiotics. Plasmid DNA from overnight cultures was purified using EZNA miniprep columns (OMEGA Bio-Tek). Plasmid sequences were verified with Sanger DNA sequencing (Eurofins Genomics).

Plasmid DNA used for all cell-free and protocell reactions was purified from EZNA midiprep columns (OMEGA Bio-Tek), followed by isopropanol and ethanol precipitation. The purified DNA pellet was reconstituted in elution buffer, measured on a Nanodrop 2000 for concentration, and stored at −20 °C.

Genomic DNA for *B. thetaiotaomicron* (ATCC 29148D) used for pathogen detection was purchased from American Type Culture Collection (ATCC). Genomic DNA for STEC O157:H7 (ATCC 51657GFP) used for pathogen detection was obtained from an overnight culture grown in tryptic soy broth supplemented with 1% glucose at 37 °C. DNA was extracted using an Invitrogen PureLink Microbiome DNA Purification Kit (A29790). Genomic DNA for DH10β and BL21 Star (DE3) were obtained from a 5 mL overnight culture grown in LB medium and were extracted using Quick-DNA Plus Kit (Zymo Research) according to the manufacturer's protocol.

**Preparation of in-house cell-free lysate.** Cellular lysate for all experiments was prepared as described by Sun et al.[62] with a few protocol modifications. Briefly, BL21 Star (DE3) Δ*lacIZYA* cells were grown in 2×YTP medium at 37 °C and 220 rpm to an optical density (OD) between 1.5–2.0, corresponding to the mid-exponential growth phase. Lysate prepared for toehold switch expression had an additional IPTG (0.4 mM) induction step when the OD reached 0.4 to activate expression of T7 RNA polymerase, creating a T7 RNAP-enriched lysate. Cells were centrifuged at 2700 × g and washed via resuspension with S30A buffer (50 mM tris, 14 mM magnesium glutamate, 60 mM potassium glutamate, 2 mM dithiothreitol, and pH-corrected to 7.7 with acetic acid). These centrifugation and wash steps were repeated twice for a total of three S30A washes. After the final centrifugation, the wet cell mass was determined, and cells were resuspended in 1 mL of S30A buffer per 1 g of wet cell mass. The cellular resuspension was divided into 1 mL aliquots. Cells were lysed using a Q125 sonicator (Qsonica) at a frequency of 20 kHz and 50% of amplitude. Cells were sonicated on ice with cycles of 10 s on and 10 s off, delivering approximately 200–250 J, at which point the cells appeared visibly lysed. An additional 4 mM dithiothreitol was added to each tube, and the sonicated mixture was then centrifuged at 12,000 × g and 4 °C for 10 min. After centrifugation, the supernatant was removed, divided into 0.5 mL aliquots, and incubated at 37 °C and 220 rpm for 80 min. After this runoff reaction, the cellular lysate was centrifuged at 12,000 × g and 4 °C for 10 min. The supernatant was removed and loaded into a 10 kDa molecular weight cutoff dialysis cassette (Thermo Fisher). Lysate was dialyzed in 1 L of S30B buffer (14 mM magnesium glutamate, 60 mM potassium glutamate, 1 mM dithiothreitol, and pH-corrected to 8.2 with tris) at 4 °C for 3 h. Dialyzed lysate was removed and centrifuged at 12,000 × g and 4 °C for 10 min. The supernatant was removed, aliquoted, flash-frozen in liquid nitrogen, and stored at −80 °C for future use.

**Cell-free reactions.** Cell-free reactions were assembled as described by Kwon and Jewett[63]. Briefly, reaction mixtures were composed of 27 v/v% of in-house prepared lysate, 2 mM each proteinogenic amino acid, 1.2 mM ATP, 0.85 mM each of GTP, CTP, and UTP, 0.2 mg/mL tRNA, 0.27 mM CoA, 0.33 mM NAD, 0.068 mM folinic acid, 1.5 mM spermidine, 33 mM PEP, 130 mM potassium glutamate, 10 mM Ammonium glutamate, 12 mM magnesium glutamate, 4 mM sodium oxalate, and specified concentrations of plasmids (described in Supplementary Table 1), RNA triggers, and small molecules. For experiments with RNA triggers (Supplementary Fig. 6), Rnase Inhibitor Murine (New England Biolabs) was added to the bulk phase at 0.5 v/v%. Each assembled cell-free reaction was 10 μL in volume and placed in a black-bottomed 384-well plate (Greiner Bio-One) and incubated at 37 °C for 3 h for GFP expression. A clear adhesive film was used to cover the plate and prevent evaporation.

**Protocell CFE reactions.** Polymers used to establish ATPS-based membrane-less protocells were prepared by dissolving either 400k Ficoll, 500k dextran, or 35k PEG

into nuclease-free water. The bulk phase at the time of preparation and before the addition of Ficoll or dextran protocells consisted of 5 v/v% of 35k PEG, 1× concentration of all reagents added for cell-free reactions (excluding lysate), specified concentrations of small molecules or nucleic acids, and nuclease-free water to a final volume of 200 μL for the 4-plex system or 100 μL for the 9-plex system. For experiments with RNA triggers (Fig. 4, Supplementary Figs. 4 and 6), Rnase Inhibitor Murine (New England Biolabs) was added to the bulk phase at 0.5 v/v%. For experiments in Fig. 5 and Supplementary Figs. 11–13, RNase Inhibitor Murine (New England Biolabs) was added to a concentration of 1.5 v/v% in the bulk phase to decrease serum Rnase activity. For colorimetric output in cell-free reactions, color substrates were added to the bulk phase to a final concentration of 0.6 mg/mL for CPRG or 0.2 mg/mL for X-gal.

Concentrations for individual plasmid sensors and reaction additives used in each figure are provided in Supplementary Table 1. Briefly, each protocell sensor consisted of 10 v/v% Ficoll or 5 v/v% dextran polymers, 27 v/v% cell-free lysate, 1× concentration of cell-free reagents, specific concentrations of plasmid DNA, and water to a final volume of 2 μL for the 4-plex system or 1 μL for the 9-plex system. The assembled protocell solution was then vortexed at a medium-high setting to ensure homogenous mixing. Protocells for detection of linear DNA also contained 2 μM of χDNA to protect linear DNA against endonucleases present in the CFE lysate, except for the Stx1 switch in Supplementary Fig. 10 which used 10 μM of the GamS protein (Arbor Bioscience). Reactions with χDNA had slightly higher fold-change compared to reactions with GamS for linear DNA protection (Supplementary Fig. 20).

To assemble the protocell arrays, a bulk phase solution containing specified concentrations of targets was first pipetted into the custom-made microwell plate (PHASIQ) to fill up each microwell. Protocell droplets were then pipetted into the bulk phase solution at their designated micro-basins. Unless otherwise noted, microwells containing assembled protocell arrays were incubated at 37 °C for 3 h before imaging on a ChemiDoc MP (Bio-Rad) imaging system. A clear adhesive film was used to cover the plate and prevent evaporation.

**Data acquisition and analysis.** For cell-free reactions, endpoint GFP readings were taken using a plate reader (Synergy4, BioTek) with its companion Gen 5 software. The excitation and emission wavelengths were 485 and 528 nm, with a gain setting of 70. For experiments in Supplementary Figs. 6–8, the gain was reduced to 60 due to signal overflow. Data acquired from Gen 5 software was exported to Microsoft Excel files for further analysis. For protocell array reactions with fluorescent reporters, the ChemiDoc MP imaging system (Bio-Rad) was used for fluorescent plate imaging. Image Lab software (Bio-Rad) was used for image collection with settings of 0.5 s exposure time, grayscale image color, 530/28 filter for GFP detection, and Blue Epi illumination as a light source. An image transform procedure was uniformly applied to all images in Image Lab (with high, low, and gamma values of 10000, 0, and 1, respectively) before exporting files for analysis. For image analysis, each image was first converted to an 8-bit grayscale image. An image processing software, Fiji, was used to manually extract signal intensities from each 1.5 mm diameter microwell (pixel size 50) for data analysis. The signal intensities were recorded and analyzed using Microsoft Excel. For colorimetric protocell array reactions, all pictures were taken with an iPhone X (Apple) in a light-controlled setting. Adobe Photoshop was used to crop individual wells for data presentation. A brightness adjustment was uniformly applied to all colorimetric ATPS photos to make them better resemble appearance to the naked eye. Untouched photos are provided with Source Data.

**Trigger preparation.** DNA encoding each trigger RNA used in experiments was either amplified from cloned plasmid or from genomic DNA of specified species of bacteria by PCR with Q5 DNA polymerase (New England Biolabs). Primers were designed to create linear DNA with a T7 promoter and additional protective sequences on the 5′ and 3′ ends of the linear template. Sequences for primers used to amplify triggers from DNA template or genomic DNA are provided in Supplementary Table 2. After PCR amplification, all products were run on a 2 w/v% agarose gel to verify successful amplification of targets and then purified using a PCR purification kit (Omega Bio-Tek). The prepared linear DNA was either directly used in cell-free and protocell reactions or used as a template for in vitro transcription.

RNA triggers were transcribed from linear DNA template using T7 polymerase according to the manufacturer's protocol (New England Biolabs). Following RNA synthesis, Dnase I (Zymo Research) was added to degrade the linear DNA template. The RNA products were then purified using an RNA Clean and Concentrator kit (Zymo Research) according to the manufacturer's protocol. Following purification, RNA concentration was measured on a Nanodrop 2000, sub-aliquoted to reduce freeze-thaw cycles, and stored −20 °C.

**STEC toehold switch development.** Toehold switches targeting gene sequences of Shiga toxin proteins (Stx1 and Stx2) in STEC O157:H7 were designed using NUPACK with series B toehold switch design[29] and cloned into a pJL1 plasmid containing a GFP reporter. Trigger sequences were synthesized by Eurofins Genomics as gene fragments containing a T7 promoter and 20–35 bp of protective sequences before and after the actual trigger sequence, and were cloned onto the pJL1 plasmid backbone to facilitate sensor screening. Trigger/sensor pairs were then tested in cell-free reactions containing 2.5 nM toehold switch sensor-GFP

reporter plasmid and 5 nM of trigger plasmid to verify successful sensor activation and orthogonality (Supplementary Fig. 21). Following trigger/sensor pair validation, primers used to amplify different trigger sequences from genomic DNA were verified for specificity toward their targets using PCR reactions with nontarget templates (Supplementary Fig. 19).

**Serum processing.** Pooled human serum was purchased from Corning (Corning, NY). Endogenous zinc was removed from serum through Chelex-100 treatment. In total, 1 g of Chelex-100 resin was added to 100 mL of serum, and the mixture was vigorously stirred for 2 h at room temperature. Resin was then isolated from the serum through centrifugation followed by syringe filtering. All serum samples were aliquoted to minimize freeze-thaw cycles and stored at −20 °C until use.

Measurement of successful zinc removal from serum was conducted at the University of Georgia Laboratory for Environmental Analysis (Supplementary Fig. 14). Samples were digested with concentrated acid and analyzed on ICP-MS according to EPA method 3052.

**Lyophilization.** Protocell sensors (1 μL) containing 55 v/v% lysate, specified concentrations of plasmid DNA (Supplementary Table 1), and 10 v/v% Ficoll were spotted into micro-basins, and the plate was stored at −80 °C to freeze for at least 5 h. Bulk phase solutions (360 μL per bulk phase condition, 120 μL per technical replicate) containing 5 v/v% PEG, 0.2 mg/mL X-gal, and 1× cell-free energy buffer were prepared in 2 mL Eppendorf tubes and stored at −80 °C to freeze for at least 5 h. Frozen plates and bulk phase solutions were transferred to a prechilled Labconco Fast-Freeze flask as quickly as possible to prevent reagent thawing. The flask was connected to a Labconco benchtop lyophilizer and lyophilized at −50 °C and 0.05 mbar overnight (>14 h).

Freeze-dried reactions were taken out of the lyophilizer the following day. For the bulk phase, water mixed with 50 nM of specified bacterial triggers was used to reconstitute the lyophilized bulk phase solution to 360 μL (120 μL per technical triplicate). One microliter of water was used to rehydrate the freeze-dried protocells, and the plate was placed at room temperature for 5 min for protocells to congeal, thereby reducing the potential for protocell sensors to float into neighboring wells during rehydration. Reconstituted bulk phase solutions were added to microwells before incubating at 37 °C. A clear adhesive film was used to cover the plate and prevent evaporation.

**Reporting summary.** Further information on research design is available in the Nature Research Reporting Summary linked to this article.

## Data availability

Source data for all main text and supplementary figures are available in the Source Data folder provided with this manuscript. Sensor sequences used are available in the Sequence Information file inside the Source Data folder provided with this manuscript. Engineering drawings of the 96- and 32-well plates used in this study are provided in the Source Data folder. Source data are provided with this paper.

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

## Acknowledgements

We thank Dr. Michael Jewett for his gift of the pJL1 plasmid. We thank Dr. Ravi Kane and his laboratory for usage of and assistance with their ChemiDoc MP imaging system. We thank Dr. Philip Santangelo and his laboratory for usage of and assistance with their lyophilizer. M.P.S. thanks the National Institutes of Health (R01EB022592) for support. S.T. thanks the National Institutes of Health (U19AI116482, R01HL136141, R01GM123517) for support.

## Author contributions

Conceptualization: Y.Z., T.K., M.P.M., M.P.S., and S.T.; Investigation: Y.Z., T.K., G.K., and M.P.M.; Formal Analysis: Y.Z. and T.K.; Writing—original draft: Y.Z.; Writing—review and editing: Y.Z., M.P.M., T.K., G.K., M.P.S., and S.T.; Visualization: Y.Z.; Supervision: M.P.S. and S.T.; Funding Acquisition: M.P.S. and S.T.

## Competing interests

The authors declare no competing interests.
