## [Peer Review File · Nature Communications]

Reviewers' Comments:

Reviewer #1:

Remarks to the Author:

Review for: Zhang Y. et al., Protocell arrays for simultaneous detection of diverse analytes

The paper by Zhang et al. describes the use of polymer-based aqueous two-phase system (i.e. PEG + Ficoll or dextran) to compartmentalize cell-free biosensors and have them operate in parallel in the same sample to perform simultaneous detection of different analytes, mainly small molecules, and nucleic acids.

The authors aim to address a major challenge in the field: many diagnostics applications rely on the use of multiple biomarkers to increase sensitivity and specificity. These biomarkers need to be detected simultaneously, and multiplexed data processing performed. An additional challenge is that there are different kinds of biomarkers: small molecules, nucleic acids, proteins. Detecting these different types of molecules simultaneously would clearly improve many diagnostics. Cell-free biosensors have been hyped for several years now as potentially capable of revolutionizing point of care diagnostics, but very few works try to actually solve challenges from the field application and get close to the real-world conditions. The authors are one of the few groups doing that, as exemplified again by this paper.

In all the question is relevant, the paper well written, and the method used convincingly. However, I still have some questions, comments, and suggestions that I think the authors should address before accepting the paper.

My main reservation concerns the number of biological replicates that does not seem adequate (see comment #7). Surprisingly to me, most experiments seem to have been performed just once (see Fig2B, 2E; 3C, 4D, 4F 5B, and 5C). And it is just plain wrong to calculate a standard deviation (and statistical significance) on technical triplicates (i.e. one experiment performed on a single day). Unfortunately, I know this is common practice in the cell-free community, but I still think it is not good. Thus I consider the paper is not acceptable without a reasonable repetition of the presented experiments.

Major comments:

1. Were replicates done for the images in Figure 6? Additionally, were any biological replicates (i.e. experiments on different days) done for any of the experiments or only technical replicates? From what I can see for bar graphs in Fig2, the authors calculate SD from three technical replicates which is just not acceptable for publication. Several experiments should be performed on different days (at least three) and the SD to the mean should be calculated.
2. The authors use the term "multiplexing" very much, although I most agree with the term parallel or simultaneous detection used in the title. Multiplexing would entails that data are integrated and processed, for example by using a smartphone software platform (something that the authors have done in previous paper). Could the authors use their previously developed smartphone platform for integrating the different data?
3. It would be interesting to measure matrix effects over different serum concentrations.
4. How did the authors select the appropriate ratios of ficoll/dextran to PEG to have "droplets" in a "bulk phase" (lines 54-55). As they are relying on this physical separation interface to segregate the components of their system, it would be nice to have a little more description of the physics for those not already familiar. Fig 2 should be modified so as describe more clearly the preparation process.
5. Additionally, the physics of this become more interesting later in the paper when the system relies on the diffusion of large molecules like DNA into the protocells, but somehow does not allow diffusion of the DNA responsible for the sensor out of the protocell. The authors should discuss this point.

6. Sometimes the reaction efficiency is better in the protocells than in single-phase, as noted in lines 208-211. The authors suggest that it is likely increased concentrations of RNA from phase separation. However, there is another plausible explanation: the protocell system contains an energy buffer that has 25x the total volume of the four protocells. This could effectively create a semi-continuous reaction that has previously been shown to increase protein production (e.g. Garamella et al. DOI: 10.1021/acssynbio.5b00296.). The authors should discuss this possibility.

7. In Figure 5B, there does not appear to be strong activation from RNA Stx1, particularly for the RNA/B12 condition. Are these two statistically significant via a post hoc test such as Tukey?

8. In lines 286-288, the authors "note that the presence of Stx1 RNA and B12 caused a slight increase in overall (background and activated) expression from the other two sensors." Assuming this refers to the +B12/Stx RNA conditions in the third part of Figure 5C, I don't notice any change in the other two sensors. Am I missing something here?

9. What was the reasoning for changing the RNA Stx1 and DNA Stx2 concentrations upon the addition of 20% Human Matrix Serum? If anything, the detection using human serum should aim to detect lower final concentrations, as the analytes in the sample would effectively be diluted 5-fold if you were measuring them endogenously. Also, how realistic are these parameters? Would you be able to detect and/or distinguish between different real world disease conditions?

Other comments:

10. What 'clear paths' are currently underway to mitigate the serum precipitation in PEG solutions noted in lines 424-425 and could you provide a reference to them? Similarly, please provide a reference for the "numerous ongoing research efforts aiming to eliminate the need for upstream input amplification" as noted in lines 434-435.

11. In lines 496-497, the authors appear to wash the cells three times for each centrifugation step. What does this wash entail? Please provide some clarification.

12. In lines 540-541, the authors state that they use 2 μM of either chi-DNA or GamS protein to protect against linear template degradation, but never state in which cases they use which. What was the reasoning for the given experiments to use one over the other?

13. Figure S5 is missing the number of replicates that were conducted.

14. Please include the concentrations of CPRG and X-gal used in Figure S8.

15. While much of the zinc from the human serum was removed from the Chelex-100 step, the measured 4 μM concentration still means that the added human serum increases the total zinc in Figure 5C by 40%. This may not be important for this proof-of-concept, but it would be worth taking that into account when further calibrating this biosensor.

16. The beginning of the introduction should be better referenced.

Reviewer #2:

Remarks to the Author:

Very recently, the number of research reports on LLPS related to biological phenomena has highly increased, and one of the most important points observed therein is that LLPS could be considered to work dynamic microcompartments which can regulate effectively temporal and spatial arrangement of biological molecules whose molecular masses are somewhat large. Interestingly, small molecules are usually able to go into and out of separated microcompartments inside cells, while large and macromolecular proteins and nucleic acids couldn't. For half a century (or more), aqueous LLPS, which is artificially generated with a binary polymer system (ATPS) like dextran/polyethylene glycol as used here, has been utilized for moderate separation of

biomacromolecules: the technology has been developed by Prof. Albertsson. Both latest LLPS and established ATPS are so interesting that similar phase segregation behavior is employed for such a similar purpose. However, in contrast to a conventional large ATPS, intracellular LLPS droplets or membraneless organelles also serve as microreactors, and from this point of view, conventional large systems have some room to be improved.

In this communication paper, by utilizing chemical micro engineering, the authors successfully constructed microdroplet systems harboring biochemical reactions operated genetically based on the ATPS, where the arrayed microdroplets served as microreactors to sense various chemical reagents; hence, the work is very interestingly because they can manage to prepare a real analogy to actual membraneless organelles. Further, in the paper, they stressed each droplet work independently (as they wrote repeatedly, "with minimal reaction crosstalk" (e.g., line 182)), but, in my opinion, crosstalk, if it occurs, is also potentially important, and they should propose some plans to use the regulated crosstalk inside the small system.

The above is my general comments and I would also like to provide a few comments on issues more specific to the results and discussion as follows:

1) As to what type of polymer could be suitable, the authors answered Ficoll compared to the case of Dextran (line 163). Is the answer significant?

2) In Figure 2, the "single protocell" and "multiplexed protocell" were compared with each other; in the latter case, droplets for empty (without plasmid) reaction seem influenced by not-empty (with plasmid) droplets. In the authors' case, and also usually, plasmid dsDNA may be partitioned into the Ficoll/Dextran droplets. Thinking of the faint signal from the empty droplet, I wonder from where GFP proteins and/or GFP-coding DNAs come to the empty rxn droplets. In addition, if such faint signals are not meaningless, this means a possibility of crosstalk observed potentially.

3) As to the interpretation on Figures 4D and 4F with Figure S5, they reported that interference of signal from each droplet was observed when the system contained trigger DNA B and H. As the author shows in Figure S5, this is very specific to combination of DNA B and H; thus, it sounds able to give more clear explanation.

4) From difference in partition coefficient about small to large molecules such as Zn^{2+} , vitamin B12, RNA and DNA, the right-most image in Figure 5B implies that RNA is not so efficiently partitioned into the droplet. I would like to hear the authors' interpretation.

5) Finally, I am curious about the stability, or the duration for which the setup can be used stably. Figures 6 and S8 show the reaction can remain entrapped for around hours, up to 18 hours (Figure S8). The phase behavior in LLPS is, of course, stable; however, since Ficoll droplets were put on the well surrounded with bulk PEG solution, that is, since the phase separation was not formed spontaneously from a binary polymer mixture, the system could seem to be metastable. If so, these droplets could be changing with time. Furthermore, the speed for changing might be affected by what reaction was entrapped. I would be happy to hear the authors' opinion.

Reviewer #3:

Remarks to the Author:

This manuscript by Zhang and colleagues describes the use of polymer-based aqueous two-phase systems to create droplets for multiplexing diagnostic sensors. Here the two-phase systems are used to partition cell-free protein expression reactions into droplets, called protocells, held within a common larger bulk phase. This is a creative idea and a nice example of using material interfaces to extend the capabilities of cell-free systems to achieve multi-mode, multiplexed gene-circuit based sensors.

The authors describe the development of the protocols for assembling membraneless protocells, evaluating parameters such as: the inclusion of energy buffer into the PEG-containing bulk phase and whether Ficoll-PEG or dextran-PEG systems provided better performance. Following

optimization of the concept, authors demonstrate the potential of the approach to provide single well, multiplexed detection of small molecules, RNA and DNA. They demonstrate the potential of the system in mock diagnostic scenarios using fluorescent and color-based outputs. Overall, it is an exciting new tool for the cell-free synthetic biology community and one that could provide meaningful improvement to the deployment of gene circuit-based diagnostics.

While authors selected candidate pathogens (*B. theta*, shiga-toxin producing *E. coli*) that might be found at high enough densities to justify nanomolar target DNA concentrations, the practical deployment of this technology in most scenarios would require an isothermal amplification step to increase the concentration of target sequences to within the detection range of toehold switches (as the authors mention in the discussion). Given the manuscript's emphasis on practical implementation, it is important to demonstrate detection of these sequences (or other sequences) over a range of concentrations to demonstrate that this concept is compatible with sensitive molecular detection. In proof-of-concept work, it would be good to see these isothermal amplification reactions run from RNA/DNA isolated from the target pathogen. While an added challenge, this is still significantly simpler than a demonstration from patient samples.

General comments

1. Manuscript lacks key experimental details throughout the figure legends. (eg. Target DNA concentrations, time points at which data were collected, replicate number, statistical method in figure legends).
2. I was also left wondering details like the concentration (%) and specific reagents (e.g. 400K Ficoll) used in the ATPSs or the type of cell-free system (e.g. in-house *E. coli* lysate-based system). These details in the main text would enrich the manuscript.
3. Microwell features in the custom 96 well plate are central to the authors being able to carry out the project. The manuscript would benefit from a description of the method used to make these microwells in the main text. CAD drawings or the information needed to replicate these plates needs to be included.
4. Page 5, line 158. Authors state "All protocells remained localized in their designated micro-basins with minimal signal crosstalk between empty and GFP-expressing reactions (Figures 2D, E), demonstrating that cell-free lysate, plasmids, and reporter proteins remain compartmentalized."

Data in figure S2 suggests that over time this may not be the case, with signals from positive and negative reactions moving toward an average signal after 4 hours (suggesting GFP diffusion between protocells).

The statement should be modified to reflect that time is an important factor for distinct signals. This phenomena also underscores why citing the time point for data collection is important in figure legends.

Do the authors have a comment on whether the observed changes (Fig. S2A) are likely the result of diffusion or protein degradation?

5. It would be good for the authors to include more aspects from the material science side of the project.
 - a. What are the material properties of Ficoll and dextran that make them good choices
 - b. How were the optimized concentrations of Ficoll and dextran determined, what concentration ranges were explored?
 - c. Is there rheological data for the material?
 - d. Given the importance of the protocells, a crisp close-up image would be beneficial
6. If trigger nucleic acids can diffuse into the protocells, how does the DNA sensor remain intact inside a specific protocell? Comment on sensor leakage and potential false positives of your technology.

7. On a related note, in the discussion the authors state "(and even concentration, for some analytes with favorable partitioning)". This phenomenon makes sense, but please reference.

8. Page 6, line 209 Authors state: "We also note that sensing reactions in protocell arrays had slightly improved limits of detection and fold-activation at low trigger concentrations compared to single-phase CFE reactions (Figure S3)". Indicate the limits of detection and quantify the fold improvement.

9. Page 8, Line 239. The statement: "Furthermore, because RNA can be readily converted into DNA using reverse-transcriptase mediated isothermal amplification techniques like Nucleic Acid Sequence-Based Amplification (NASBA) or Recombinase Polymerase Amplification (RPA)" should read "...Reverse Transcription Recombinase Polymerase Amplification (RT-RPA)"

10. Page 8, line 259. Authors state "We characterized these micronutrient biosensors together in protocell arrays at concentrations near the physiologically relevant range to confirm sensor performance (Figure S6)." Here it is important to define in the text what the authors define as the threshold for detection. These values should in the main text, alongside healthy and pathological concentrations of B12 and Zinc.

11. Page 9, line 284. Authors state "We spiked 20% human serum (similar to sample concentrations used in our previously published CFE efforts)⁶ into the bulk phase and observed the same target specific activation from all protocell sensors (Figure 5C),...". As written, this could be misunderstood as identical performance as the experiment from Figure 5B. Sensors response shown in Figure 5C is to a higher concentration of analytes and, in some cases, is different (pink bars in the two right experiments).

Similarly, in figure 5C, the experiment is run in 20% serum. Would it be realistic to expect the related concentrations of Zinc and B12 in patients (5x concentration tested)?

12. Page 9, line 289. Define the term "matrix effect" for the readers. This will provide clarity on the phenomenon for readers looking at the figure, as well as a connection to the related text in the discussion.

Page 13, line 423. "While serum protein precipitation did not obstruct biphasic polymer separation or detection of GFP signal, it hindered pigment visualization in colorimetric reactions (data not shown)." If this text is referring to the matrix effect, it would be useful to cite Figure 5C.

13. Page 10, Line 327. How much longer? Specify time for each.

14. Page 10, line 332. Authors state "Within three hours, all protocell sensors turned visibly blue to accurately report on the presence of the triggers in the bulk phase, with minimal pigment leakage or sensor crosstalk (Figure 6B)." The nature of these triggers should be clearly stated in the text. In this case, after looking at the methods section, it appears as though the triggers are genomic material from the related organisms. This information adds relevance to the data and makes for a more informative read.

15. Page 12, line 395: Authors state: "...sub-systems can all be used together in parallel to measure different targets in the same sample; of note, the protocells sensing nucleic acids in Figure 5 use different lysate preparations than the other protocells in Figure 5." This is an interesting and relevant detail, but, as written, the information is not clearly conveyed. A little more specific detail would be helpful.

Figure specific comments.

Fig 2B. Key, relevant experimental details should be stated in figure captions and the body of text.

- a) State incubation temperature
- b) State time point at which data was collected
- c) Clarify whether the triplicates are from three different wells or three different spots in a single well
- d) State plate reader measurement parameters (eg. Ex/Em wavelength)

Fig 2E. The phrase "empty reaction" may be confusing to the reader and it may be more clear to refer to control reactions as "No DNA". As written, these data points could be understood to empty reactions without the cell-free system components.

Fig 3. Specify concentration of DNA sensors

Fig 4A. The trigger RNA is drawn to include a hairpin structure. This is not a consistently present feature of trigger RNA and may be best left out.

Fig 4D. Include the time point used for data collection

Fig S7A. Figure legend does not reference the sensors used in the experiment

Fig 5BC. Labeling of components on left side of figure panel C is not included.

We thank the reviewers for their thorough, thoughtful consideration of our work. We have done a substantial number of additional experiments and have made substantive changes to the text to address the reviewers' requests and suggestions. The additional work has confirmed the robustness of our results and improved the manuscript overall, making it more compelling. All of the comments from the reviewers are included below; we offer point-by-point responses to all of them, indicating the work we have done and the changes we have made to accommodate each request. Original reviewer comments are in plain font, and our responses are in indented blue font.

Reviewer #1 (Remarks to the Author):

Review for: Zhang Y. et al., Protocell arrays for simultaneous detection of diverse analytes

The paper by Zhang et al. describes the use of polymer-based aqueous two-phase system (i.e. PEG + Ficoll or dextran) to compartmentalize cell-free biosensors and have them operate in parallel in the same sample to perform simultaneous detection of different analytes, mainly small molecules, and nucleic acids.

The authors aim to address a major challenge in the field: many diagnostics applications rely on the use of multiple biomarkers to increase sensitivity and specificity. These biomarkers need to be detected simultaneously, and multiplexed data processing performed. An additional challenge is that there are different kinds of biomarkers: small molecules, nucleic acids, proteins. Detecting these different types of molecules simultaneously would clearly improve many diagnostics. Cell-free biosensors have been hyped for several years now as potentially capable of revolutionizing point of care diagnostics, but very few works try to actually solve challenges from the field application and get close to the real-world conditions. The authors are one of the few groups doing that, as exemplified again by this paper.

In all the question is relevant, the paper well written, and the method used convincingly. However, I still have some questions, comments, and suggestions that I think the authors should address before accepting the paper.

My main reservation concerns the number of biological replicates that does not seem adequate (see comment #7). Surprisingly to me, most experiments seem to have been performed just once (see Fig2B, 2E; 3C, 4D, 4F 5B, and 5C). And it is just plain wrong to calculate a standard deviation (and statistical significance) on technical triplicates (i.e. one experiment performed on a single day). Unfortunately, I know this is common practice in the cell-free community, but I still think it is not good. Thus I consider the paper is not acceptable without a reasonable repetition of the presented experiments.

We thank the reviewer for the kind comments and appreciation of our work, and we have reproduced essentially all figures per the request of the reviewer.

The definition of the appropriate type of replication for these experiments is a non-trivial task. Certainly, all of the work described had already been replicated in-house in the iterations over the course of the research, and the results initially presented were representative of the different experiments. In writing this paper, we believed that reaction replicates were a reasonable representation of the sample-to-sample variability since these chemical systems (while complex, not defined) do not truly have "biological

variability”, and differences between days are often the result of systematic error (temperature fluctuations in incubators, pipetting errors in master mixes, etc.) that are not directly indicative of the technology. In many cases, our primary goal is to demonstrate overall behavior and trends rather than day-to-day reproducibility of absolute readings.

Nonetheless, we appreciate the reviewer’s point, their scientific rigor, and their desire for demonstration of research reproducibility – particularly in the context of potential diagnostic applications. It is indeed important that the literature is reproducible, and reproducibility is certainly as much an issue in the cell-free field as it is elsewhere (if not more so). Accordingly, we have included biological replicates for all figures presented in the main text and for figures in the supplementary information that are used to draw or support any quantitative conclusion. Only figures S9, S10, and S15 remain with only technical triplicates, since S9 and S10 were used only for tuning, and in these figures the biological replicates of sensor and substrate behaviors have been replicated in other figures.

Major comments:

1. Were replicates done for the images in Figure 6? Additionally, were any biological replicates (i.e. experiments on different days) done for any of the experiments or only technical replicates? From what I can see for bar graphs in Fig2, the authors calculate SD from three technical replicates which is just not acceptable for publication. Several experiments should be performed on different days (at least three) and the SD to the mean should be calculated.

We thank the reviewer for this additional elaboration from the initial overarching comment. As described above, we now include 3 biological replicates, each with 3 technical replicates, for all the images shown in Figure 6 (see Figure S16 for fresh reactions and Figure S17 for lyophilized reactions). We have done the same for all figures in the main text and all figures in the supplementary information that are used to draw or support any quantitative conclusion; only S9, S10, and S15 are excluded, for reasons indicated above. Each biological replicate represents an independently assembled reaction on a different day. Each technical replicate represents a separate reaction derived from the same master mix. Standard deviations are now calculated based on at least 9 replicates (3 biological replicates x 3 technical replicates, although individual plots may use more replicates as indicated in figure captions), as we believe this reasonably captured the variance in the data points to give a more conservative estimation of the true error.

Selecting the method to define the error bars for this experimental design was non-trivial, even after consultation with a biostatistician. While we have seen literature reports where error bars are based on, e.g., the three means of experiments from different days that each consist of three technical replicates, we believe this does not accurately capture variation. The mean of three technical replicates will always be a more consistent value than any individual measurement, and so using three means to calculate a standard deviation (relative to the mean of those three means) was, in our opinion, too likely to lead to an underestimation of overall variability. Since we were plotting standard deviations (rather than the standard error of the mean), including more replicates would not artificially decrease our error bars. Accordingly, for all figures, error

bars are based on all replicates (as described above), even when main text figures only plot the means of different days' experiments for the sake of legibility.

2. The authors use the term "multiplexing" very much, although I most agree with the term parallel or simultaneous detection used in the title. Multiplexing would entails that data are integrated and processed, for example by using a smartphone software platform (something that the authors have done in previous paper). Could the authors use their previously developed smartphone platform for integrating the different data?

We appreciate the reviewer's suggestion. In a general sense, we do perform the "integration and processing" that the reviewer first alludes to; all sensors use a single output channel (fluorescence or colorimetric output), with their signals encoded via spatial positioning that is then processed downstream by the bench scientist (either by quantifying results in imaging software or via naked eye interpretation of different colorimetric outputs). Additionally, the term "multiplexing" is often used in the literature (including in a number of our cited references) for the type of work we described here. However, we appreciate that different readers may have different standards or opinions about what technically constitutes "multiplexing". Accordingly, we have changed "multiplexing" to simultaneous or parallel analyte detection in many locations (approximately 80% of occurrences) throughout the text, reserving its use for only when most appropriate or necessary for clarity.

We also thank the reviewer for their appreciation of our smartphone software platform from our previous publication. Unfortunately, the previous software development is not directly generalizable due to considerable differences in assay format, particularly in the size of our protocell droplets (1 μ L) vs. pelleted cells pooled from 5 mL of culture, as well as in the use of fluorescence here vs. colorimetric readouts in our previous work. Our previously developed software platform is not designed to handle the protocell array's data size and format, and re-engineering would be difficult and beyond the scope of this proof-of-concept work.

3. It would be interesting to measure matrix effects over different serum concentrations.
Matrix effects are an ongoing topic of interest in our group, and identifying the impact of different serum concentrations on protocell arrays is indeed interesting and adds value to this manuscript. We have now included a 10% serum condition in the supplementary information (Figure S12). We could not test serum concentrations beyond 20% because protein precipitation in the PEG phase obstructed micro-basin visualization during reaction assembly. In general, addition of serum in the bulk phase negatively impacts CFE protein production due to nuclease activities in serum (Ref 5, 37, 38), consistent with our previous work (Figure 1e in Ref 5). To reach the same terminal output with the same inducer concentration, sensor concentrations had to be increased with increasing serum concentrations (Figures S11-13, Table S1), demonstrating the matrix effects that the reviewer brought up.
4. How did the authors select the appropriate ratios of ficoll/dextran to PEG to have "droplets" in a "bulk phase" (lines 54-55). As they are relying on this physical separation interface to segregate the components of their system, it would be nice to have a little more description of the physics for those not already familiar. Fig 2 should be modified so as describe more clearly the preparation process.

The reviewer raises an excellent point. Appropriate polymer ratios for biphasic separation were selected based on our previous binodal characterization of the two aqueous two-phase systems (Ref 25, 27). Biphasic separations of PEG-Ficoll and PEG-dextran polymers are spontaneous processes based on polymer interactions (Ref 17, 19, 25, 27). The configuration of droplets in a bulk phase is achieved by simply pipetting a denser phase solution (Ficoll/ dextran) droplet into a less dense bulk phase created by PEG (Figure S2a). We have updated our introduction (lines 50-60) as well as in the results (lines 125-128) and in Figures 2 and S2 to provide better background for a more general audience and to better reflect polymer ratio selection, biomolecule partitioning, and protocell array preparation methods.

5. Additionally, the physics of this become more interesting later in the paper when the system relies on the diffusion of large molecules like DNA into the protocells, but somehow does not allow diffusion of the DNA responsible for the sensor out of the protocell. The authors should discuss this point.

We thank the reviewer for this suggestion. Compartmentalization of the machinery and nucleic acids inside these membrane-less protocells relies primarily on partitioning; while diffusion is the physical mechanism by which the DNA makes its way from the bulk phase to the protocells, what keeps it there are thermodynamic driving forces (enthalpic and entropic considerations for equilibrium) that make diffusion of nucleic acid polymers out of the protocell energetically unfavorable. This partitioning-based compartmentalization is similar to the situation for membrane-less organelles of eukaryotic cells but contrasts with membrane-bound organelles where diffusion out of the organelle by certain molecules would be thermodynamically favorable were it not for a physical barrier composed of lipid bilayers. We additionally note that by utilizing favorable partitioning (rather than a physical barrier) to compartmentalize nucleic acids, membrane-less protocells have the advantage that there is not a physical barrier for the nucleic acids to enter the compartment. Thus, thermodynamically-driven nucleic acid partitioning drives and maintains selective compartmentalization of biological macromolecules of interest in the droplet phases of the protocells (Ref 16, 20, 21, and 24). DNA compartmentalization in ATPSs has been reported by others for PEG-dextran and ficoll ATPS (for example, Ref 20), as well as in coacervates (Ref 24). We have updated our introduction to provide more of this detailed explanation as well as more references to partitioning behavior (lines 56-60).

6. Sometimes the reaction efficiency is better in the protocells than in single-phase, as noted in lines 208-211. The authors suggest that it is likely increased concentrations of RNA from phase separation. However, there is another plausible explanation: the protocell system contains an energy buffer that has 25x the total volume of the four protocells. This could effectively create a semi-continuous reaction that has previously been shown to increase protein production (e.g. Garamella et al. DOI: 10.1021/acssynbio.5b00296.). The authors should discuss this possibility.

The reviewer raises an interesting point. In our replication efforts, we identified a miscalculation of RNA concentration in the CFE reaction. After correcting our error, we did not observe a significant difference in the fold activation between toehold switch sensors in our protocell array vs. a typical CFE reaction (Figure S6, S7). Nevertheless, we do acknowledge the unique reaction format of our protocell array has similarities to a CFE reaction operating in dialysis mode, which is reported to have enhanced protein yield by prolonging CFE reactions (Ref 56), as suggested by the reviewer. We have

discussed the advantages of dialysis-mode operation in the discussion and referenced relevant literature (lines 474-479).

7. In Figure 5B, there does not appear to be strong activation from RNA Stx1, particularly for the RNA/B12 condition. Are these two statistically significant via a post hoc test such as Tukey?

We thank the reviewer for this observation. Different switches have different limits of detection and dynamic ranges, and our selection of Stx1 concentration in the initial draft was perhaps suboptimal for demonstrating the protocell technology. In performing the requested replication experiments, we increased the RNA concentration for Stx1 to 0.5 μM for all replicates in Figure 5b and observed stronger Stx1 sensor activation. This also better aligned the RNA concentration in the water matrix (Figure 5b) with that used in the serum matrix (Figure 5c), also addressing a later question by the reviewer.

8. In lines 286-288, the authors "note that the presence of Stx1 RNA and B12 caused a slight increase in overall (background and activated) expression from the other two sensors." Assuming this refers to the +B12/Stx RNA conditions in the third part of Figure 5C, I don't notice any change in the other two sensors. Am I missing something here?

We apologize for this miscommunication; while the reviewer's quoted text came immediately after we mentioned Figure 5c, we actually were referring to the behavior in Figure 5b. This should have been worded better. At any rate, this phenomenon was observed to a lesser extent in the serum matrix in the original manuscript, but was still evident in the zinc sensor even in the serum matrix. We have adjusted the main text to better explain this phenomenon (lines 349-352). In both Figure 5b-c and S11-13, the activation for zinc sensor is higher in the "+ all" condition compared to the "+ zinc & + DNA Stx2" condition. The basal level expression of the zinc sensor is also higher in the "+ B12 & + RNA Stx1" condition compared to the no inducer condition. Hence, we hypothesize that matrix effects from the addition of vitamin B12 and Stx1 RNA caused a gain of background signal in the zinc sensing protocell. No consistent increase in signal is observed for the Stx2 sensor.

9. What was the reasoning for changing the RNA Stx1 and DNA Stx2 concentrations upon the addition of 20% Human Matrix Serum? If anything, the detection using human serum should aim to detect lower final concentrations, as the analytes in the sample would effectively be diluted 5-fold if you were measuring them endogenously. Also, how realistic are these parameters? Would you be able to detect and/or distinguish between different real world disease conditions?

The reviewer raises some important issues. The decision to increase RNA Stx1 and DNA Stx2 in 20% human serum was due the inhibitory effects of serum matrix on protein production (Ref 5). Serum matrix contains numerous nucleases (Ref 37-38) that can degrade double-stranded DNA (like plasmid sensors and DNA Stx2 trigger) and RNA (such as the RNA Stx1 trigger and mRNA transcripts made in protocell reactions), even when reactions are supplemented with RNase inhibitor. We originally increased target concentrations to provide comparable signal to the no-serum case, but in retrospect we should have just kept those concentrations constant and more explicitly noted the known potential decrease of expression for cell-free reactions in serum. In this revision, we have now kept the analyte concentrations constant across 0-20% serum in the bulk phase and adjusted sensor concentration to counteract serum's matrix effect instead. We have adjusted the main text to reflect this change (lines 344-346, 356-361).

On the reality of sensing analyte at the presented concentrations, zinc is detected at clinically relevant concentrations – the threshold for zinc deficiency is in the range of 8.5 - 11.5 μM , corresponding to 1.7 - 2.3 μM in 20% human serum matrix (Ref 5, 57). Unfortunately, there is no CFE sensor sensitive enough to detect clinically relevant concentrations of vitamin B12 (15 - 80 pM in 20% human serum) (Ref 4, 58). Thus, we aimed to demonstrate vitamin B12 sensor activation in human serum as a proof-of-principle. We have now explicitly discussed these clinically relevant ranges and associated limitations in the text at lines 495-499. Regarding the detection of Stx1 and Stx2 triggers, it indeed is impractical to find RNA and DNA in serum at concentrations of 500 nM and 75 nM, respectively. In the Discussion section, we have expanded upon our previous mention of the possibility of using nucleic acid amplification techniques, such as Nucleic Acid Sequence-Based Amplification (NASBA) or Recombinase Polymerase Amplification (RPA) to bring targets typically present in the femtomolar range to the nanomolar response range of our toehold switches (lines 501-505). In addition, we have further demonstrated our platform's compatibility with RPA as an initial proof-of-principle (Figure S18), demonstrating the feasibility of using amplification techniques to detect nucleic acid analytes at clinically relevant concentrations. We did not demonstrate multiplexed amplification of trigger targets or the addition of RPA reaction directly in serum matrix, as the extensive optimization of primer pairs and reaction conditions required was beyond the scope of our proof-of-principle work.

Other comments:

10. What 'clear paths' are currently underway to mitigate the serum precipitation in PEG solutions noted in lines 424-425 and could you provide a reference to them? Similarly, please provide a reference for the "numerous ongoing research efforts aiming to eliminate the need for upstream input amplification" as noted in lines 434-435.

We thank the reviewer for these excellent suggestions. We have provided an explicit description and a reference for possibility to address the serum precipitation issue (lines 492-494): using clear bottomed micro-wells (Ref 17). We have also provided a reference toward amplification-free detection nucleic acids (Ref 59, line 511).

11. In lines 496-497, the authors appear to wash the cells three times for each centrifugation step. What does this wash entail? Please provide some clarification.

We apologize for this mistake and thank the reviewer for pointing it out; we incorrectly described the protocol. What we meant to say is that the cells were washed in S30A buffer *via* resuspension and then centrifuged to remove the S30A buffer. This process was repeated two times for a total of three washes. We have updated the methods to reflect this change (lines 566-569). This washing step is common in *E. coli* lysate preparation, though the number of washes vary in different published protocols (Krinsky *et al.* doi:10.1371/journal.pone.0165137).

12. In lines 540-541, the authors state that they use 2 μM of either chi-DNA or GamS protein to protect against linear template degradation, but never state in which cases they use which. What was the reasoning for the given experiments to use one over the other?

We apologize for any ambiguity we left on this point in the initial draft's main text. While individual reaction conditions (CFE lysate used, plasmid concentrations, and reaction additives) are provided in Table S1, we have referenced this information more

prominently in the main text, captions, and supplementary figures, and have directly addressed this question from the reviewer in the Methods (lines 615-619).

As to the reviewer's point: in the revised manuscript, all relevant figures use 2 μM χDNA protein to protect linear DNA except for the Stx1 switch in Figure S10 and Figure S20, which has 10 μM GamS. We generally use these interchangeably but have tried to be more consistent in this revision; we left Figure S10 as it was since it was only used for tuning of concentrations, and the results from the selected concentrations were subsequently recapitulated with χDNA in the main text figures. We also did further assessment on GamS protein and χDNA , showing similar protection ability offered by both nuclease inhibitors with no apparent reason to prefer one over the other (Figure S20).

We also note that we made a typographical error in the first submission; it should be 10 μM GamS, not 2 μM . This has been corrected (line 617).

13. Figure S5 is missing the number of replicates that were conducted.

We apologize for this oversight. Original Figure S5, now Figure S8 has been updated to indicate the (new) number of replicates (3 biological replicate x 3 technical replicate).

14. Please include the concentrations of CPRG and X-gal used in Figure S8.

We apologize for this oversight. We have included these concentrations (0.6 mg/mL for CPRG and 0.2 mg/mL for X-gal) in original Figure S8, now Figure S15.

15. While much of the zinc from the human serum was removed from the Chelex-100 step, the measured 4 μM concentration still means that the added human serum increases the total zinc in Figure 5C by 40%. This may not be important for this proof-of-concept, but it would be worth taking that into account when further calibrating this biosensor.

We thank the reviewer for pointing this out. We have accounted for the remaining zinc present in human serum in this revision (Figure 5c, S12-13).

16. The beginning of the introduction should be better referenced.

We thank the reviewer for this suggestion. We have added more references in the first paragraph of the introduction (lines 17-25).

Reviewer #2 (Remarks to the Author):

Very recently, the number of research reports on LLPS related to biological phenomena has highly increased, and one of the most important points observed therein is that LLPS could be considered to work dynamic microcompartments which can regulate effectively temporal and spatial arrangement of biological molecules whose molecular masses are somewhat large. Interestingly, small molecules are usually able to go into and out of separated microcompartments inside cells, while large and macromolecular proteins and nucleic acids couldn't. For half a century (or more), aqueous LLPS, which is artificially generated with a binary polymer system (ATPS) like dextran/polyethylene glycol as used here, has been utilized for moderate separation of biomacromolecules: the technology has been developed by Prof. Albertsson. Both latest LLPS and established ATPS are so interesting that similar phase segregation behavior is employed for such a similar purpose. However, in contrast to a conventional large ATPS, intracellular LLPS droplets or membraneless organelles also serve as microreactors, and from this point of view, conventional large systems have some room to be improved.

In this communication paper, by utilizing chemical micro engineering, the authors successfully constructed microdroplet systems harboring biochemical reactions operated genetically based on the ATPS, where the arrayed microdroplets served as microreactors to sense various chemical reagents; hence, the work is very interestingly because they can manage to prepare a real analogy to actual membraneless organelles. Further, in the paper, they stressed each droplet work independently (as they wrote repeatedly, "with minimal reaction crosstalk" (e.g., line 182)), but, in my opinion, crosstalk, if it occurs, is also potentially important, and they should propose some plans to use the regulated crosstalk inside the small system.

We thank the reviewer for the kind comments and appreciation of our work. Regarding crosstalk (by which we mean off-target effects specific to a particular sensor), we did not observe significant crosstalk at the sensor level (CFE machinery, plasmids, and macromolecular reaction additives stay compartmentalized relatively well inside the Ficoll protocell in the first 3 hours). However, we did observe signal interference from the target analytes, such as the inhibition phenomenon specific to triggers B & H (Figure 4 and S8) and matrix effects discussed in the main text (lines 349-352, Figure 5). We also note that, when one protocell is a high signal producer in the microwell, the GFP fluorescence may diffuse to the surrounding micro-basin, causing an overall increase in the GFP background (Figures 2-5).

While in a multiplexed sensor context we often worry about crosstalk as a problem, the reviewer seems to be highlighting an interesting possibility for our protocell array technology: to actually *use* crosstalk (perhaps intentionally created or regulated) to enable new types of experiments. We have explicitly noted this exciting possibility in the discussion section (lines 468-474).

The above is my general comments and I would also like to provide a few comments on issues more specific to the results and discussion as follows:

- 1) As to what type of polymer could be suitable, the authors answered Ficoll compared to the case of Dextran (line 163). Is the answer significant?

The direct answer to the reviewer's question is: we do not believe the fact that Ficoll was better than dextran has any broader scientific significance beyond it being the optimal choice for this engineering application. In further protocell characterization done for this revision, we found that protocells created by 10% Ficoll had broad compatibility with differently prepared CFE lysates (Figure S2b-c), which was desirable for potential sensing applications. In contrast, protocells formed by 10% dextran strongly inhibited protein synthesis in some CFE lysates. While 5% dextran protocells supported protein synthesis across CFE lysates, these protocells began to disintegrate at 3 hours (Figures 2c-d, S2d-e). In comparison, 10% Ficoll had longer stability (Figure 2, Figure S2). The main text and supplementary information have been adjusted to reflect this additional information (lines 125-128, 168-171). Ultimately, though, this was just an engineering optimization from a small sampling of parameter space, and other formulations could work as well as, or better than, the one we demonstrated in our work.

- 2) In Figure 2, the "single protocell" and "multiplexed protocell" were compared with each other; in the latter case, droplets for empty (without plasmid) reaction seem influenced by not-empty (with plasmid) droplets. In the authors' case, and also usually, plasmid dsDNA may be partitioned into the Ficol/Dextran droplets. Thinking of the faint signal from the empty droplet, I wonder from where GFP proteins and/or GFP-coding DNAs come to the empty rxn droplets. In addition, if such faint signals are not meaningless, this means a possibility of crosstalk observed potentially.

The reviewer raises an excellent point. Based on examination of the fluorescence images (Figures S2d-e, 2c), the background signal in the empty droplets is likely due to either degradation of certain types of protocells or diffusion of GFP from a high GFP-producing protocell to the surrounding bulk phase. In our previous ELISA work, we observed that the formulation of polymer ATPS strongly influenced protein retention (Figure 2 of Ref 19). Since the current formulation was selected to optimize protein expression efficiency rather than protein retention, it is possible that at very high levels of GFP expression, a sufficient amount of GFP partitions to the bulk phase to be detectable. From there, it may either stay in the bulk phase or partition into empty protocells. This does indeed amount to some degree of non-specific crosstalk based only on protocell location rather than the actual sensor in the protocell. To more fully mitigate this issue, further optimization of polymer ratios would be needed. We have adjusted the main text to acknowledge GFP diffusion in the protocell array format (lines 173-178).

- 3) As to the interpretation on Figures 4D and 4F with Figure S5, they reported that interference of signal from each droplet was observed when the system contained trigger DNA B and H. As the author shows in Figure S5, this is very specific to combination of DNA B and H; thus, it sounds able to give more clear explanation.

The reviewer is right, and we apologize for not giving a more clear statement of this. We have attempted to clarify this point in the main text (lines 238-242).

- 4) From difference in partition coefficient about small to large molecules such as Zn²⁺, vitamin B12, RNA and DNA, the right-most image in Figure 5B implies that RNA is not so efficiently partitioned into the droplet. I would like to hear the authors' interpretation.

We apologize for any confusion that our initial experimental design choices may have caused. Different toehold switches have different limits of detection, so the specific concentrations of target sequence added for RNA vs. DNA are not directly comparable

across different switches. Based on this point, and in response to a comment from Reviewer 1, we have increased the concentration for Stx1 trigger RNA in this revision to show stronger switch activation (Figure 5b, Figure S11).

- 5) Finally, I am curious about the stability, or the duration for which the setup can be used stably. Figures 6 and S8 show the reaction can remain entrapped for around hours, up to 18 hours (Figure S8). The phase behavior in LLPS is, of course, stable; however, since Ficoll droplets were put on the well surrounded with bulk PEG solution, that is, since the phase separation was not formed spontaneously from a binary polymer mixture, the system could seem to be metastable. If so, these droplets could be changing with time. Furthermore, the speed for changing might be affected by what reaction was entrapped. I would be happy to hear the authors' opinion.

Our time course characterization in Figure S2d-e corroborates the reviewer's comment of a potentially metastable protocell array system. We showed that ATPS formed by 10% Ficoll and 5% PEG start to show diffusion of output signal beyond 4 hours, and all the detection experiments throughout the manuscript were done at 3 hours. There are two main reasons we generally use formulations including just one of the two phase-forming polymers in the droplet phase. One is that we have previously demonstrated that use of non-pre-equilibrated ATPS systems can accelerate reactions due to convection driven by the metastable system (ref 25). The other is that when we dehydrate the droplet phase, it often forms cleaner and longer-lasting dried phases when PEG is not included. We have now added these discussions in the main text (lines 121-124).

Reviewer #3 (Remarks to the Author):

This manuscript by Zhang and colleagues describes the use of polymer-based aqueous two-phase systems to create droplets for multiplexing diagnostic sensors. Here the two-phase systems are used to partition cell-free protein expression reactions into droplets, called protocells, held within a common larger bulk phase. This is a creative idea and a nice example of using material interfaces to extend the capabilities of cell-free systems to achieve multi-mode, multiplexed gene-circuit based sensors.

The authors describe the development of the protocols for assembling membraneless protocells, evaluating parameters such as: the inclusion of energy buffer into the PEG-containing bulk phase and whether Ficoll-PEG or dextran-PEG systems provided better performance. Following optimization of the concept, authors demonstrate the potential of the approach to provide single well, multiplexed detection of small molecules, RNA and DNA. They demonstrate the potential of the system in mock diagnostic scenarios using fluorescent and color-based outputs. Overall, it is an exciting new tool for the cell-free synthetic biology community and one that could provide meaningful improvement to the deployment of gene circuit-based diagnostics.

While authors selected candidate pathogens (*B. theta*, shiga-toxin producing *E. coli*) that might be found at high enough densities to justify nanomolar target DNA concentrations, the practical deployment of this technology in most scenarios would require an isothermal amplification step to increase the concentration of target sequences to within the detection range of toehold switches (as the authors mention in the discussion). Given the manuscript's emphasis on practical implementation, it is important to demonstrate detection of these sequences (or other sequences) over a range of concentrations to demonstrate that this concept is compatible with sensitive molecular detection. In proof-of-concept work, it would be good to see these isothermal amplification reactions run from RNA/DNA isolated from the target pathogen. While an added challenge, this is still significantly simpler than a demonstration from patient samples.

We thank the reviewer for the kind comments and appreciation of our work.

We also acknowledge the importance and relevance of demonstrating compatibility with workflows likely to be important for field applications. To this end, we have now demonstrated the protocell array's compatibility with an isothermal nucleic acid amplification technique (RPA) by showing successful detection of bacterial pathogen triggers amplified by RPA (Figure S18).

General comments

1. Manuscript lacks key experimental details throughout the figure legends. (eg. Target DNA concentrations, time points at which data were collected, replicate number, statistical method in figure legends).

We apologize for this oversight. We have updated our figure legends throughout to include key experimental details suggested by the reviewer, and have also more prominently referenced Table S1 that contains all of the experimental details for each figure.

2. I was also left wondering details like the concentration (%) and specific reagents (e.g. 400K Ficoll) used in the ATPSs or the type of cell-free system (e.g. in-house *E. coli* lysate-based system). These details in the main text would enrich the manuscript.
*Again, we apologize for this oversight. We have now included details on the polymer concentrations and reagents (lines 119-121) and explicitly stated that the system is an in-house *E. coli* lysate-based system (lines 95-96 and 118).*
3. Microwell features in the custom 96 well plate are central to the authors being able to carry out the project. The manuscript would benefit from a description of the method used to make these microwells in the main text. CAD drawings or the information needed to replicate these plates needs to be included.
We appreciate the reviewer's point here. We include the CAD drawing for the 9 microbasin, 32 well plate and a PDF drawing of the 4 microbasin 96 well plate. We now include these files to supplement the images of the plates in Figure S1.
4. Page 5, line 158. Authors state "All protocells remained localized in their designated microbasins with minimal signal crosstalk between empty and GFP-expressing reactions (Figures 2D, E), demonstrating that cell-free lysate, plasmids, and reporter proteins remain compartmentalized.". Data in figure S2 suggests that over time this may not be the case, with signals from positive and negative reactions moving toward an average signal after 4 hours (suggesting GFP diffusion between protocells).

The statement should be modified to reflect that time is an important factor for distinct signals.

We thank the reviewer for this suggestion. We have adjusted the discussion in this section to reflect the metastable nature of our protocell array system (lines 168-178, Figure S2).

This phenomena also underscores why citing the time point for data collection is important in figure legends.

The reviewer raises an excellent point. We have now included time point for data collection in all figure legends.

Do the authors have a comment on whether the observed changes (Fig. S2A) are likely the result of diffusion or protein degradation?

*We do believe these phenomena are likely the result of protein diffusion similar to what we have observed previously for ELISA (Figure 2 of Ref 19). As for protein degradation, we don't think this would be significant in our system as the strain BL21 Star (DE3) used for lysate preparation belongs to the *E. coli* B strain and is deficient in two main proteases OmpT and Lon (Rosano *et al.* doi.org/10.1002/pro.3668), and protein degradation ought not lead to background fluorescence signal.*

5. It would be good for the authors to include more aspects from the material science side of the project.
We thank the reviewer for this helpful overall suggestion. We will address each specifically requested aspect below.
 - a. What are the material properties of Ficoll and dextran that make them good choices

Ficoll and dextran are selected based on their ability to form stable phase-separation with PEG over a range of concentrations, their ability to form stable films in the dried state, and for their ability to accelerate reactions during rehydration of the dried droplet phase (Ref 19, 25, 27). These polymers are also selected based on their compatibility with CFE systems and previously-demonstrated ability to compartmentalize CFE reactions (Ref 20, 24). In the main text, we have explicitly stated these characteristics in the Results section on lines 122-125, and we have referenced more studies on polymer ATPS properties in the main text (lines 56-60, 67-72).

- b. How were the optimized concentrations of Ficoll and dextran determined, what concentration ranges were explored?

Concentrations of Ficoll and dextran were selected based on previously characterized binodal curves for PEG-Ficoll and PEG-dextran system (Refs 25, 27). We have now referenced our previous characterizations in the main text (lines 125-126). In this work, dextran polymer was tested at 5% and 10% in protocell reactions. Ficoll polymer was only tested at 10% and PEG was only tested at 5%. Characterizations supporting the selection of Ficoll based on its suitability for multiple different types of CFE lysate preparations are provided in Figure S2b-c. We did not attempt extensive optimization of concentrations.

- c. Is there rheological data for the material?

We have referenced more detailed characterizations of PEG-Ficoll and PEG-dextran aqueous two phase system from our previous works (Ref 25, 27) in text (line 128). While no quantitative measures have been made, the specific Ficoll system is less viscous than the specific dextran system we used.

- d. Given the importance of the protocells, a crisp close-up image would be beneficial
We have included microscopy images of PEG-Ficoll and PEG-dextran protocells in Figure S2a.

6. If trigger nucleic acids can diffuse into the protocells, how does the DNA sensor remain intact inside a specific protocell? Comment on sensor leakage and potential false positives of your technology.

The reviewer raises a good point, also noted by Reviewer #1. We copy below our response to that question:

Compartmentalization of the machinery and nucleic acids inside these membrane-less protocells relies primarily on partitioning; while diffusion is the physical mechanism by which the DNA makes its way from the bulk phase to the protocells, what keeps it there are thermodynamic driving forces (enthalpic and entropic considerations for equilibrium) that make diffusion of nucleic acid polymers out of the protocell energetically unfavorable. This partitioning-based compartmentalization is similar to the situation for membrane-less organelles of eukaryotic cells but contrasts with membrane-bound organelles where diffusion out of the organelle by certain molecules would be thermodynamically favorable were it not for a physical barrier composed of lipid bilayers. We additionally note that by utilizing favorable partitioning (rather than a physical barrier) to compartmentalize nucleic acids, membrane-less protocells have the advantage that there is not a physical barrier for the nucleic acids to enter the compartment. Thus, thermodynamically-driven nucleic acid partitioning drives and maintains selective

compartmentalization of biological macromolecules of interest in the droplet phases of the protocells (Ref 16, 20, 21, and 24). DNA compartmentalization in ATPSs has been reported by others for PEG-dextran and ficoll ATPS (for example, Ref 20), as well as in coacervates (Ref 24). We have updated our introduction to provide more of this detailed explanation as well as more references to partitioning behavior (lines 56-60).

7. On a related note, in the discussion the authors state “(and even concentration, for some analytes with favorable partitioning)”. This phenomenon makes sense, but please reference. This is an excellent suggestion. We have included references (line 440, Ref 16, 20, 49) in the discussion.
8. Page 6, line 209 Authors state: “We also note that sensing reactions in protocell arrays had slightly improved limits of detection and fold-activation at low trigger concentrations compared to single-phase CFE reactions (Figure S3)”. Indicate the limits of detection and quantify the fold improvement.
We thank the reviewer for this suggestion. We have indicated the limits of detection and fold activation in the supplementary figures comparing CFE and protocell array sensitivity (now Figures S6-7). In our replication efforts, we identified a miscalculation of RNA concentration in the CFE reaction. After correcting our error, we did not observe a significant difference in the fold activation between toehold switch sensors in our protocell array vs. a typical CFE reaction, so we have amended the text to eliminate this content.
9. Page 8, Line 239. The statement: “Furthermore, because RNA can be readily converted into DNA using reverse-transcriptase mediated isothermal amplification techniques like Nucleic Acid Sequence-Based Amplification (NASBA) or Recombinase Polymerase Amplification (RPA)” should read “...Reverse Transcription Recombinase Polymerase Amplification (RT-RPA)”
We apologize for this mistake; we have updated the main text to reflect this change (lines 231-232).
10. Page 8, line 259. Authors state “We characterized these micronutrient biosensors together in protocell arrays at concentrations near the physiologically relevant range to confirm sensor performance (Figure S6).” Here it is important to define in the text what the authors define as the threshold for detection. These values should in the main text, alongside healthy and pathological concentrations of B12 and Zinc.
The reviewer makes an excellent suggestion. We have changed the specific text referred to in this critique, and instead discuss the relationship to clinically relevant concentrations in the discussion, in part in response to another reviewer’s request (lines 495-499).
11. Page 9, line 284. Authors state “We spiked 20% human serum (similar to sample concentrations used in our previously published CFE efforts)⁶ into the bulk phase and observed the same target specific activation from all protocell sensors (Figure 5C),...”. As written, this could be misunderstood as identical performance as the experiment from Figure 5B. Sensors response shown in Figure 5C is to a higher concentration of analytes and, in some cases, is different (pink bars in the two right experiments).
The reviewer raises an excellent point here, as did another reviewer. We copy below part of a response from above:

The decision to increase RNA Stx1 and DNA Stx2 in 20% human serum was due the inhibitory effects of serum matrix on protein production (Ref 5). Serum matrix contains numerous nucleases (Ref 37-38) that can degrade double-stranded DNA (like plasmid sensors and DNA Stx2 trigger) and RNA (such as the RNA Stx1 trigger and mRNA transcripts made in protocell reactions), even when reactions are supplemented with RNase inhibitor. We originally increased target concentrations to provide comparable signal to the no-serum case, but in retrospect we should have just kept those concentrations constant and more explicitly noted the known potential decrease of expression for cell-free reactions in serum. In this revision, we have now kept the analyte concentrations constant across 0-20% serum in the bulk phase and adjusted sensor concentration to counteract serum's matrix effect instead. We have adjusted the main text to reflect this change (lines 344-346, 356-361).

We have also adjusted the main text to try to address the specific concern raised by this reviewer by removing the phrase "the same" (lines 344-346).

12. Similarly, in figure 5C, the experiment is run in 20% serum. Would it be realistic to expect the related concentrations of Zinc and B12 in patients (5x concentration tested)?

The reviewer raises an excellent point also brought up by another reviewer. We copy below the relevant response from above:

On the reality of sensing analyte at the presented concentrations, zinc is detected at clinically relevant concentrations – the threshold for zinc deficiency is in the range of 8.5 - 11.5 μM , corresponding to 1.7 - 2.3 μM in 20% human serum matrix (Ref 5, 57). Unfortunately, there is no CFE sensor sensitive enough to detect clinically relevant concentrations of vitamin B12 (15 - 80 pM in 20% human serum) (Ref 4, 58). Thus, we aimed to demonstrate vitamin B12 sensor activation in human serum as a proof-of-principle. We have now explicitly discussed these clinically relevant ranges and associated limitations in the text at lines 495-499.

13. Page 9, line 289. Define the term "matrix effect" for the readers. This will provide clarity on the phenomenon for readers looking at the figure, as well as a connection to the related text in the discussion.

We apologize for this oversight. We have updated the main text to more explicitly define and explain matrix effects (lines 352-355).

Page 13, line 423. "While serum protein precipitation did not obstruct biphasic polymer separation or detection of GFP signal, it hindered pigment visualization in colorimetric reactions (data not shown)." If this text is referring to the matrix effect, it would be useful to cite Figure 5C.

This statement was not referring to matrix effects (as it did not affect quantitative readouts), but instead a different phenomenon where serum proteins seemed to form cloudy precipitates when added to PEG. We have modified the relevant text in an attempt towards more clarity.

14. Page 10, Line 327. How much longer? Specify time for each.

This is a good point; we have explicitly stated the additional time needed for X-gal cleavage (30 minutes, line 375).

15. Page 10, line 332. Authors state “Within three hours, all protocell sensors turned visibly blue to accurately report on the presence of the triggers in the bulk phase, with minimal pigment leakage or sensor crosstalk (Figure 6B).” The nature of these triggers should be clearly stated in the text. In this case, after looking at the methods section, it appears as though the triggers are genomic material from the related organisms. This information adds relevance to the data and makes for a more informative read.

We apologize for the ambiguity; we have updated the main text to explicitly state this (lines 292, 384-385).

16. Page 12, line 395: Authors state: “...sub-systems can all be used together in parallel to measure different targets in the same sample; of note, the protocells sensing nucleic acids in Figure 5 use different lysate preparations than the other protocells in Figure 5.” This is an interesting and relevant detail, but, as written, the information is not clearly conveyed. A little more specific detail would be helpful.

This is a great suggestion; we have added more detail and discussion in the main text (lines 304-313, 452-457) to better convey this feature of our protocell array.

Figure specific comments.

We appreciate the reviewer’s attention to detail; we have addressed all of the figure-specific requests, and have noted below any other relevant information.

Fig 2B. Key, relevant experimental details should be stated in figure captions and the body of text.

- a) State incubation temperature
- b) State time point at which data was collected
- c) Clarify whether the triplicates are from three different wells or three different spots in a single well
- d) State plate reader measurement parameters (eg. Ex/Em wavelength)

We have included all information requested in the Figure 2 caption (lines 143-147).

The main text was also updated to include incubation time and temperature (lines 132, 166-167). Plate reader and imager measurement parameters are provided in the Methods (lines 628-630 and 632-634, respectively).

Fig 2E. The phrase “empty reaction” may be confusing to the reader and it may be more clear to refer to control reactions as “No DNA”. As written, these data points could be understood to empty reactions without the cell-free system components.

This is a very helpful suggestion. We have updated legends for empty reactions “No DNA” for all of our figures.

Fig 3. Specify concentration of DNA sensors

We have included in the concentration of plasmid sensors directly in the circuit diagram in Figure 3a.

Fig 4A. The trigger RNA is drawn to include a hairpin structure. This is not a consistently present feature of trigger RNA and may be best left out.

We have removed the 5’ hairpin drawn in Figure 4a.

Fig 4D. Include the time point used for data collection

We have included the time point and incubation temperatures (3 hours at 37 °C) in all figure legends.

Fig S7A. Figure legend does not reference the sensors used in the experiment.

We apologize for this mistake. Figure legend for original Figure S7, now Figure S10, has been updated.

Fig 5BC. Labeling of components on left side of figure panel C is not included.

We apologize for this mistake. We have included the label for panel c.

Reviewers' Comments:

Reviewer #1:

Remarks to the Author:

I thank the authors for their answers, the main points of concern and questions have been addressed. The manuscript now appears stronger and clearer and is acceptable for publication in Nature Communications.

Reviewer #2:

Remarks to the Author:

Carefully considering the reviewers' comments to the original version, the authors have apparently made efforts to improve the results and discussion based on solid experimental data and more robust logics in the revised manuscript. As to the queries that I had raised to the original manuscript, the authors have given the clear answers with appropriate amendments. I would be happy that their addition for the revision according to my comments could help readers to understand their work. Hence, I have not had any further query on this present version.

Reviewer #3:

Remarks to the Author:

Overall authors were responsive and provided the requested edits and additions to the manuscript. Thank you. It is an exciting report and one that will catalyze new ideas for the field.

Response to Reviewers:

Reviewers' comments are reproduced verbatim in black, our responses are indented and in blue.

Reviewer #1 (Remarks to the Author):

I thank the authors for their answers, the main points of concern and questions have been addressed. The manuscript now appears stronger and clearer and is acceptable for publication in Nature Communications.

We thank the reviewer for another thorough review of our manuscript and appreciate the reviewer's contributions to this improved work. We are delighted that the reviewer is satisfied with our revision. Thank you for recommending our work for publication in Nature Communications.

Reviewer #2 (Remarks to the Author):

Carefully considering the reviewers' comments to the original version, the authors have apparently made efforts to improve the results and discussion based on solid experimental data and more robust logics in the revised manuscript. As to the queries that I had raised to the original manuscript, the authors have given the clear answers with appropriate amendments. I would be happy that their addition for the revision according to my comments could help readers to understand their work. Hence, I have not had any further query on this present version.

We thank the reviewer for another thorough review of our manuscript and appreciate the reviewer's contributions to this improved work. We are delighted that the reviewer is satisfied with our revision. Thank you for recommending our work for publication in Nature Communications.

Reviewer #3 (Remarks to the Author):

Overall authors were responsive and provided the requested edits and additions to the manuscript. Thank you. It is an exciting report and one that will catalyze new ideas for the field.

We thank the reviewer for another thorough review of our manuscript and appreciate the reviewer's contributions to this improved work. We are delighted that the reviewer is satisfied with our revision and finds our work exciting for the field. Thank you for recommending our work for publication in Nature Communications.

Editorial Requests:

All editorial requests have been addressed. The checklist provided with the last decision has been included in the following pages.